



# Some effects of flow expansion on the aerodynamics of horizontal-axis wind turbines

David H. Wood[1] and Eric J. Limacher[2]

[1]Department of Mechanical and Manufacturing Engineering, University of Calgary, Calgary T2N 1N4, AB, Canada.
[2]Department of Mechanical Engineering, Federal University of Pará, Belém, Brazil

**Correspondence:** David H. Wood (dhwood@ucalgary.ca)

**Abstract.** Upwind of an energy-extracting horizontal-axis wind turbine, the flow expands as it approaches the rotor, and the expansion continues in the vorticity-bearing wake behind the rotor. The upwind expansion has long been known to influence the axial momentum equation through the axial component of the pressure, although the extent of the influence has not been quantified. Starting with the impulse analysis of Limacher & Wood (2020), but making no further use of impulse techniques,
we demonstrate that the expansion redistributes momentum from the external flow to the wake and derive its exact expression when the rotor is circumferentially uniform. This expression, which depends on the radial velocity and the axial induction factor, is added to the thrust equation containing the pressure on the back of the disk. Removing the pressure to obtain a practically useful equation shows the axial induction in the far-wake is twice the value at the rotor only at high tip speed ratio and only if the relationship between vortex pitch and axial induction in non-expanding flow carries over to the expanding case.
At high tip speed ratio, we assume that the expanding wake approaches the "Joukowsky" model of a hub vortex on the axis of rotation and tip vortices originating from each blade. The additional assumption that the helical tip vortices have constant pitch, allows a semi-analytic treatment of their effect on the rotor flow. Expansion modifies the relation between the pitch and induced axial velocity so that the far-wake area and induction are significantly less than twice the values at the rotor. There is a moderate decrease – about 6% – in the power production and a similar size error occurs in the familiar axial momentum
equation involving the axial velocity.

## 1   Introduction

Conservation of axial and angular momentum are fundamental principles for wind turbine analysis. They are applied using control volumes (CVs) such as those in figure 1, or more commonly, to a CV coinciding with a mean streamtube and extending into the far-wake, the hypothetical region of no further wake development. For blade-element momentum theory, the CVs
become expanding annular streamtubes intersecting the elements. The change in axial or angular momentum of the flow determines the net thrust or torque, respectively, acting on the rotor or blade elements, e.g. Burton et al. (2011), Hansen (2015), and Sørensen (2016). Angular momentum is easier to analyze because in most cases it is generated only at the blades.

When a turbine extracts kinetic energy from the wind, the flow must expand both upwind and downwind of the rotor. As noted on p. 185 of Glauert (1935), and by Goorjian (1972), the axial momentum equation may receive contributions from the





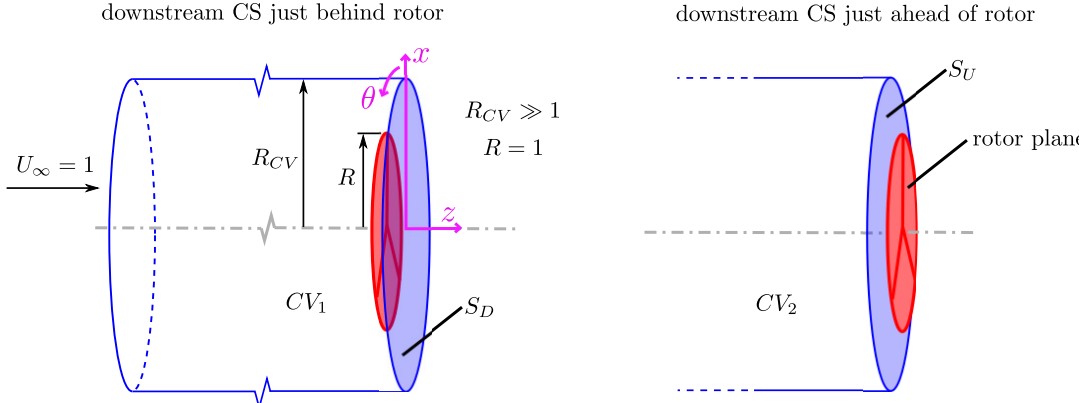

**Figure 1.** Control volumes (CVs) to be used in the present analysis. In both variants, the upstream face extends in $z$ to $-\infty$, where the velocity is the wind speed, and $R_{CV} \gg R$. The downstream control surface is just downstream and just upstream of the rotor plane in $CV_1$ and $CV_2$, respectively, and the corresponding donwstream control surfaces (CS) are labelled $S_D$ and $S_U$. Taken from LW.

pressure in the expanding flow upwind of the rotor. The expansion causes the pressure force to have an axial component which must be redistributive; that is, the total rotor thrust is not changed but momentum can be redistributed from one part of the flow to another. This is because the pressure forces acting on the cylindrical control surfaces at radius $R_{CV}$ in figure 1 are entirely radial. Although redistribution has been recognized for a long time, and is discussed by Sørensen (2016) and van Kuik (2018) among others, a satisfactory analysis of it is lacking. The first main result of the present analysis a closed-form expression for

the redistributive pressure force for a circumferentially uniform rotor.

Limacher & Wood (2020) (hereinafter "LW") investigated steady wind turbine thrust, $T$, using an impulse analysis, whereby the pressure in the axial momentum equation for any CV is replaced by terms that include vorticity fluxes across the CV boundaries. We will use what we call the "impulse perspective" as explained below, but not impulse techniques in this paper; the interested reader is referred to LW for a short history and more details. LW showed that by approximating a rotor as an

actuator disk, $T$ is given exactly by integration over the face $S_D$ of $CV_1$, situated just downwind of the rotor on the left of figure 1:

$$\frac{T}{\rho} = \int\limits_{S_D} \left( \frac{1}{2} w^2 + \lambda w x \right) dS. \tag{1}$$

where $\rho$ is the air density, and $w$ is the circumferential velocity on $S_D$; in LW, $w$ denoted the circumferential velocity at the rotor plane, which was assumed to be one half that on $S_D$. $\lambda$ is the tip speed ratio, and $x$ is the radius normalized by the tip

radius so that $x \leq 1$ for the rotor. The downwind face of the second CV in the figure, $S_U$, is just upwind of the rotor. The term "exact" will be used throughout this paper to indicate that no assumptions beyond those listed below have been invoked. Taking the "wake" to be the flow that has passed through the rotor, these assumptions are:



1. the flow upwind of the rotor and outside the wake, is inviscid, steady, and spatially uniform, and the blades rotate at constant angular velocity,

2. the total energy of the wake is reduced instantaneously at the rotor, after which it is conserved,

3. viscous and/or Reynolds stresses can be neglected on the CV surfaces,

4. the axial, $u$, and radial velocity, $v$, are continuous through the rotor disk,

5. viscous drag is negligible,

6. $w$ is zero in the upwind flow and outside the wake,

7. the vorticity in the wake is concentrated in line vortices or vortex sheets aligned with the local streamlines in the rotating frame of reference. In other words, the wake vortices rotate rigidly with the blades and vortex lines and streamlines coincide, and

8. to derive the exact blade element equation which is the differential form of Equation (1), the vorticity piercing the lateral boundaries of the annular CVs intersecting the blade elements must have no effect on the element's thrust.

Assumption #7 simplifies the terms involving the trailing vorticity crossing $S_D$ in the impulse derivation. Assumptions #3, #5, and #8 are likewise embedded in the equations derived by LW, and are not explicitly reinvoked in the analysis to follow. We note, emphatically, that none of the eight assumptions places any restrictions on flow expansion. Since the impulse derivation of (1) is likewise unrestricted, the equation is exact in the presence of flow expansion.

Although Equation (1) has been known since Glauert (1935), and appears in modern texts, such as Equation (4.6) in van
Kuik (2018), LW's analysis provides the first proof of its exactness when the trailing vortex sheets have finite thickness. LW's second main finding for circumferentially-uniform, expanding flow, is

$$0 = \int_{S_U} \left(v^2 - a^2\right) dS = \int_{S_D} \left(v^2 - a^2\right) dS \tag{2}$$

where $a = 1 - u$ is the usual axial induction factor. van Kuik (2020) found Equation (2) was satisfied by his model of the expanding flow through a wind turbine rotor. When $a$ and $v$ are further assumed to be $C_0$-continuous on $S_U$ and $S_D$, Equation
(2) tells us that $|a| = |v|$ at some radial location, and LW cite three simulations that show $|a| \approx |v|$ near the rotor tip. The vanishing of the first integral on $S_U$ in (2) is the more general result; the vanishing of the second integral on $S_D$ follows from assumption #4 above. Until the end of Section 3, we treat the rotor as circumferentially uniform. Since Equation (1) contains no terms representing pressure redistribution, LW assert that its blade element version is also exact:

$$\frac{1}{\rho}\frac{dT}{dx} = \int_{0}^{2\pi} \left(\frac{1}{2}w^2 + \lambda wx\right) x\, d\theta = 2\pi wx \left(\frac{w}{2} + \lambda x\right), \tag{3}$$

where $\theta$ is the circumferential co-ordinate. This result is also not new: it is, for example, Equation (4.24) of van Kuik (2018). It is often referred to as the "Kutta-Joukowsky" theorem for blade element thrust because it gives the axial force, $dT/dx$, as





the product of the circumferential velocity *at the rotor*, $w/2 + \lambda x$, and the sum of the circulation on all blades, $2\pi wx$. Impulse analysis, however, can also be applied if the CV outlet is moved to the far-wake to give $dT/dx$ in terms of the $w$ in the far-wake. Glauert's (1935) original derivation of Equation (1) – based on the Bernoulli equation – also suggests the exactness of (3).

Equations (19) and (24) of Okulov & Sørensen (2008), express the helical symmetry of the flow immediately behind the rotor containing trailing vortices of constant pitch, $p$, and constant radius:

$$\frac{p}{x} = \frac{w/2}{a} = \frac{1-a}{w/2 + \lambda x}. \tag{4}$$

Combining Equations (3) and (4) gives

$$\frac{1}{\pi\rho}\frac{dT}{dx} = \left(w^2 + 2\lambda wx\right)x = 4\left(1-u\right)ux = 4\left(1-a\right)ax \tag{5}$$

recovering the conventional axial momentum equation. In Section 4 we show that (4) is altered by flow expansion which reduces the accuracy of the conventional axial momentum equation.

When $\lambda = 0$ the flow expansion is minimal and Equation (5) is accurate from which it follows that $w^2 \approx 4a$; that is, the induced circumferential velocity is significantly larger than the induced axial velocity. We use this result in Section 3 to examine the axial momentum equation when $\lambda = 0$.

Equation (2) can be derived using standard CV momentum analysis, but the authors are unaware of it appearing in the literature prior to LW. It is a natural outcome of the impulse perspective which we use to investigate the effects of flow expansion on the conventional axial momentum equation. It will be shown that (2) is closely related to the effects of pressure in the upwind flow on the conventional axial momentum equation, and the general relationship between $a$ and the far-wake induction, $a_\infty$. $T$ is derived in Chapter 4 of Sørensen (2016) and Section 5.2.4 of van Kuik (2018) using a CV ending in
the far-wake. We take the different approach of using the CVs shown in Figure 1 because that choice clarifies the effects of expansion. We also make further use of the impulse form of the $T$ equation. The derivations of the remaining equations in this paper are straightforward, and could have been easily done in the past if the impulse perspective had been available.

The next Section describes the first derivation of the redistribution term for rotor thrust when the flow is circumferentially uniform, and the subsequent section analyzes the blade element form of the thrust equation. It contains our second main result,
that $a_\infty \approx 2a$ is possible only at high $\lambda$ and *only if* Equation (4) remains valid for expanding flow. In Section 4, we apply the Biot-Savart law to an expanding Joukowsky wake which contains only hub and tip vortices. On the further assumption of constant $p$, we show, again for the first time, that $a \leq a_\infty \leq 2a$. Not surprisingly, the far-wake radius is reduced as is the power extracted by the turbine. The final two sections contain the general discussion and conclusions, respectively.

## 2  Rotor thrust for expanding flow

Some results of the impulse analysis can be converted easily into conventional equations containing the axial velocity and the pressure on the CV surface even when the flow expands through the rotor. For example, Bernoulli's equation for $P_U$, the





pressure on $S_U$, is

$$\frac{2P_U}{\rho} = 1 - v^2 - u^2 \tag{6}$$

Equation (6) allows the removal of $v^2$ from (2) to give

$$\int_0^\infty \frac{P_U}{\rho} x dx = \int_0^\infty u(1-u) x dx \tag{7}$$

which is also the outcome of a conventional momentum balance on $CV_2$. The momentum balance on $CV_1$ yields

$$\frac{T}{2\pi\rho} = \int_0^\infty u(1-u) x dx - \int_0^\infty \frac{P_D}{\rho} x dx. \tag{8}$$

where $P_D$ is the pressure on $S_D$. It is important to note that the effective upper limit on the integrals in (8) is outside the wake. Nevertheless,

$$\frac{T}{2\pi\rho} = \int_0^1 \frac{P_U - P_D}{\rho} x dx = \int_0^1 \frac{\Delta P}{\rho} x dx \tag{9}$$

Since $P_D = P_U$ for $x > 1$. The thrust equation with integration only over the rotor, can be found by rewriting (8) as

$$\frac{T}{2\pi\rho} = \int_0^1 a(1-a) x dx - \int_0^1 \frac{P_D}{\rho} x dx + \int_1^\infty a(1-a) x dx - \int_1^\infty \frac{P_D}{\rho} x dx. \tag{10}$$

To remove the last two integrals for $x \geq 1$, we use Equation (6) for $P_D = P_U$ and then (2), to arrive at

$$\frac{T}{\pi\rho} = 2\int_0^1 a(1-a) x dx - 2\int_0^1 \frac{P_D}{\rho} x dx + \int_0^1 (a^2 - v^2) x dx. \tag{11}$$

The first integral in (11) contributes half the conventional thrust. It and the second integral are components of conventional CV analysis, whereas the third integral is new. It makes (11) exact for an actuator disk when the flow expands and is redistributive because, as noted above, the integral of $v^2 - a^2$ over the whole of $S_D$ is zero. The last integral in (11), however, is shown below to be generally positive. We now change the CV from that shown in figure 1 to the more commonly-used one formed by the bounding streamsurface ($BS$) dividing the flow passing through the rotor from the external flow. $BS$ begins at $z = -\infty$ where $z$ is the axial co-ordinate with origin at the rotor, figure 1. The vertical faces of the new CV are, therefore, subsets of those shown in figure 1. A straightforward momentum balance gives

$$\frac{T}{\pi\rho} = 2\int_0^1 a(1-a) x dx - 2\int_0^1 \frac{P_D}{\rho} x dx + 2\int_{-\infty}^0 \left( P\frac{dx}{dz} \right)_{BS} x dz \tag{12}$$

where the last integrand is evaluated on $BS$. It follows immediately from (11) and (12) that

$$\int_{-\infty}^0 \left( P\frac{dx}{dz} \right)_{BS} x dz = \frac{1}{2} \int_0^1 (a^2 - v^2) x dx \tag{13}$$





which gives the first quantification known to the authors of the axial force due to the expanding flow through a wind turbine rotor. It is easy to generalize this equation. For any $x$ and $z \leq 0$:

$$\int\limits_{-\infty}^{z} \left( P\frac{dx}{dz} \right)_{S(x,z)} x dz = \frac{1}{2} \int\limits_{0}^{x} \left( a^2 - v^2 \right) x dx \qquad (14)$$

where $S(x,z)$ is the streamsurface passing through $(x,z)$ so that $BS = S(1,0)$. The second integral is evaluated at $z \in [-\infty, 0]$.

$P_D$ in (11) can be evaluated in the standard manner by assuming that the unsteady Bernoulli equation is valid from immedi-

130 ately behind the rotor to the far-wake:

$$-\frac{2P_D}{\rho} = u^2 + v^2 + w^2 - u_\infty^2 - w_\infty^2 - \frac{2P_\infty}{\rho} - 2\lambda x_\infty w_\infty + 2\lambda x w \qquad (15)$$

where the far-wake terms have the subscript "$\infty$". The last two terms arise from the unsteady potential terms, evaluated by assuming rigid wake rotation (see appendix B of LW). Conveniently, these terms cancel due to conservation of angular momentum, yielding

$$-\frac{2P_D}{\rho} = u^2 + v^2 + w^2 - u_\infty^2 - w_\infty^2 - \frac{2P_\infty}{\rho}. \qquad (16)$$

Combining (11) and (16) we get

$$\frac{T}{\pi\rho} = \int\limits_{0}^{1} \left( 1 - u_\infty^2 \right) x dx - \int\limits_{0}^{1} \left( \frac{2P_\infty}{\rho} - w^2 + w_\infty^2 \right) x dx, \qquad (17)$$

where $w_\infty$ and $P_\infty$ are evaluated at $x_\infty$ in the wake, connected to $x$ at the rotor by a mean streamsurface.

In the far-wake, the pressure and circumferential velocity are related by

140 $$\frac{dP_\infty/\rho}{dx} = \frac{w_\infty^2}{x}. \qquad (18)$$

The relationship between the area integrals of $P$ and $w$ can be found using the technique introduced by McCutchen (1985) and rediscovered by Wood (2007): multiply both sides by $x^2$ and integrate, by parts for the left side. If $P_\infty x^2 \to 0$ as $x \downarrow 0$, and is zero at the edge of the far-wake, then

$$\int\limits_{0}^{R_\infty} \frac{P_\infty}{\rho} x dx = -\frac{1}{2} \int\limits_{0}^{R_\infty} w_\infty^2 x dx. \qquad (19)$$

As pointed out by van Kuik (2018) in conjunction with his Equation (6.8), any swirl at the edge of the wake makes $P_\infty(x_\infty) \neq 0$. The present analysis can accommodate this behaviour but for the present we take the simpler path of assuming $P_\infty(x_\infty) = 0$. The main justification for this assumption is that we expect the magnitude of the swirl to become negligible everywhere at the edge of the wake at high $\lambda$. When $P_\infty(x_\infty) = 0$, Equation (17) reduces to

$$\frac{T}{\pi\rho} \approx \int\limits_{0}^{1} \left( 1 - u_\infty^2 \right) x dx + \int\limits_{0}^{1} w^2 x dx, \qquad (20)$$



Defining the axial induction in the far wake as $a_\infty = 1 - u_\infty$, we obtain

$$\frac{T}{\pi \rho} \approx \int\limits_0^1 a_\infty \left(2 - a_\infty\right) x \, dx + \int\limits_0^1 w^2 x \, dx, \tag{21}$$

and the standard thrust equation is recovered if $a_\infty \approx 2a$ and $w^2 \approx 0$ which is typically the case at high $\lambda$ but may not be generally correct. Note that (21) is accurate at $\lambda = 0$ where the first integral is negligible but $a_\infty \neq 2a$.

To recover the classical thrust equation, and to provide a comparison to the analyses of Sørensen (2016) and van Kuik (2018),

we now move the downwind face of the CV to the far-wake and use Equation (19). This results in

$$\frac{T}{2\pi \rho} = \int\limits_0^{R_\infty} a_\infty \left(1 - a_\infty\right) x \, dx - \int\limits_0^{R_\infty} \frac{P_\infty}{\rho} x \, dx = \int\limits_0^{R_\infty} a_\infty \left(1 - a_\infty\right) x \, dx + \frac{1}{2} \int\limits_0^{R_\infty} w_\infty^2 x \, dx. \tag{22}$$

If we ignore the second integral in (17) and the integrals in (22) containing $P$ and $w$, and assume $a$ and $a_\infty$ are constant with $x$, we again recover the conventional relation $a_\infty \approx 2a$ once the conservation of mass is invoked.

In considering the blade element equation for $dT/dx$ in the next Section, it is useful to have the alternative form of (22) from

the impulse analysis of LW. Assuming that $v = 0$ everywhere in the far-wake and $a_\infty = 0$ for $r > R_\infty$ gives

$$\frac{T}{2\pi \rho} = \int\limits_0^{R_\infty} \left(\frac{1}{2} w_\infty^2 + \lambda w_\infty x\right) x \, dx - \frac{1}{2} \int\limits_0^{R_\infty} a_\infty^2 x \, dx \tag{23}$$

from their Equation (22), whose general form we write as

$$\frac{T}{2\pi \rho} = \int\limits_0^\infty \left(\frac{1}{2} w^2 + \lambda w x\right) x \, dx + \frac{1}{2} \int\limits_0^\infty \left(v^2 - a^2\right) x \, dx. \tag{24}$$

This equation holds anywhere behind the rotor, i.e. for $z > 0$ with the second integral approaching zero as $z \to 0$.

## 3   Blade element thrust in expanding flow

It is easy to show that the blade element form of (11),

$$\frac{1}{\pi \rho} \frac{dT}{dx} = \left[2a\left(1 - a\right) - \frac{2P_D}{\rho} + a^2 - v^2\right] x, \tag{25}$$

is exact for a circumferentially-uniform rotor in expanding flow. This can be done in at least two ways. First, by simple manipulation, the bracketed terms become

$1 - u^2 - v^2 - \dfrac{2P_D}{\rho} = \dfrac{2\Delta P}{\rho}$ \hfill (26)

and the pressure difference across the annulus containing the blade elements must give the exact thrust by assumption #4. Secondly, starting from Equation (14) it is easy to prove that $a^2 - v^2$ in (25) accounts for the difference in pressure acting





on the top and bottom of the expanding annular streamtube that intersects the blade elements. Equation (2) shows that the $a^2 - v^2$ term in (25) has the necessary property of being redistributive, but we have shown for the first time that redistribution

of momentum occurs over the entire face $S_D$, and not just the rotor.

We now consider the consequences of the exact Equation (25) for the far-wake. If the $w$ and $P_\infty$ terms in (16) are negligible at high $\lambda$, the bracketed term in (25) becomes

$$2a(1-a) - \frac{2P_D}{\rho} + a^2 - v^2 \approx 1 - u_\infty^2. \qquad (27)$$

The exactness of the blade element form of (22) is not easy to establish in general because all three velocity components can

be important in the wake and the total pressure is not constant. This is the first reason we based our analysis on the CVs shown in figure 1 rather than one extending to the far-wake. We note, however, that there is no redistribution in the flow outside the far-wake where $v^2 = a^2 = 0$. In other words, the redistribution of momentum by the pressure is completed before the far-wake is reached. This is the second reason we used the CVs in figure 1. Further, it is reasonable, but unproven, to assume that redistribution is complete everywhere within the wake. Then, the blade element form of (22) will have a term corresponding

to the bracketed term in (25) of $2u(1 - u_\infty)$. Combining with (27), we retrieve the standard result that $u = (1 + u_\infty)/2$ or $a_\infty = 2a$ which can be accurate only at high $\lambda$; note that the discussion immediately below (21) shows the result does not hold at $\lambda = 0$. Further, from (6)

$$\frac{2P_U}{\rho} = 1 - u^2 - v^2 = 2a(1-a) + a^2 - v^2. \qquad (28)$$

If Equation (5) is valid, then (25) becomes

$$-\frac{2P_D}{\rho} \approx 2a(1-a) - a^2 + v^2. \qquad (29)$$

for $x \leq 1$. Assuming $P_U = -P_D$ gives $a = v$, which cannot be correct in general for several reasons. First, $v \to 0$ as $x \downarrow 0$ whereas there is no similar constraint on $a$. Secondly, we argued above that pressure redistribution occurs partly outside the wake so $a \neq v$ for all $x > 1$ or Equation (2) would be violated if $a = v$ for $x \leq 1$. Thirdly, van Kuik (2018) Section 5.4.4 points out that there is no theoretical requirement that $P_U = -P_D$. They are unlikely, however, to differ greatly in general. This

suggests $v \to a$ as $x \to 1$, as argued by LW, and shown by the model calculations of van Kuik (2020), who found also that $v$ was significantly larger than $a$ outside the wake until at least $x \approx 1.2$. If $a > v$ over most of the rotor, then the positive $a^2 - v^2$ in (25) represents a redistribution of momentum from the external flow to the wake.

A more definite statement about $P_U$ and $P_D$ can be made for stationary rotors where (5) is accurate, because flow expansion is small enough to be neglected, but not exact, because flow expansion is neglected. This requires $w^2 \approx 4a$ and terms in $a^2$ and

200 $v^2$ can be neglected. so that

$$\frac{P_U}{\rho} \approx -\frac{P_D}{\rho} \approx a \qquad (30)$$

and any subsequent inequality as $\lambda$ increases is due to nonzero $a^2 - v^2$.





We now consider the far-wake in more detail to determine the vortex pitch and its relation to $a_\infty$ which are required in the next section. Equation (18) requires $P_\infty/\rho = -w_\infty^2/2$ when $w_\infty \sim 1/x_\infty$. We assume that at sufficiently high $\lambda$, the flow

downwind of the rotor approximates a Joukowsky wake with the hub vortex lying along the axis of rotation and the tip vortices at radius $R_\infty$ in the far-wake, with no vorticity in between. The main justification for this assumption comes from Equation (1). When $\lambda = 0$, the first term implies that the bound vorticity, $\Gamma$, cannot be constant; Wood (2015) showed that $\Gamma \sim x^2$. At high $\lambda$ however, the first term becomes negligible in comparison to the second for most $x$. The simplest wake for which the thrust remains bounded on a turbine with $N$ blades occurs when $N\Gamma\lambda \sim \lambda w x$ is constant in $x$ and $\lambda$; this is the Joukowsky

wake in which Assumption #7 of Section 1 becomes irrelevant to the flow between the tip and hub vortices. Further, the tip vortices now separate the wake and the external flow which may have very different velocities. The vortex velocity should then be the average of these two and the vortex lines need not align with the wake streamlines.

Outside the hub vortex core of a Joukowsky wake, $w_\infty \sim 1/x_\infty$ and, as pointed out by Sørensen (2016), the total pressure is constant for all streamsurfaces. In addition, blade element independence will hold in the sense that the integrands in (22) and

(23) must be equal. Thus

$$\frac{1}{2}w_\infty^2 + \lambda w_\infty x - \frac{1}{2}a_\infty^2 = a_\infty\left(1 - a_\infty\right) + \frac{1}{2}w_\infty^2 \tag{31}$$

without making any assumption about the relationship between $a$ and $a_\infty$. $p_\infty$, the constant pitch of the constant radius tip vortices, is related to the velocities by $p_\infty/x_\infty = w_\infty/a_\infty$ and (31) can be rewritten as

$$p_\infty = \frac{1 - a_\infty/2}{\lambda}. \tag{32}$$

In the next section, $\lambda$ will be calculated using Equation (32) for a given $p_\infty$ and the corresponding calculated value of $a_\infty$. Equation (32) is the high$-\lambda$ equivalent of Equation (22) of Okulov & Sørensen (2008) for vortex pitch provided the convection velocity of the vortex—$w$ in their notation but $w_v$ here—is equal to $a_\infty/2$. Table 1 of Wood & Okulov (2017) shows that $w_v \to a$ for ideal Betz-Goldstein rotors as $\lambda \to \infty$ and so (32) is recovered since $a_\infty \to 2a$ in the same limit. Another way to view this result is that the axial velocity in the Joukowsky far-wake is constant and equal to $1 - a_\infty$ outside the vortex cores,

so the tip vortices must travel downwind at a velocity of $1 - a_\infty/2$ to be force-free.

The Kawada-Hardin (KH) equations, Kawada (1936), Hardin (1982), for a doubly-infinite helical vortex of constant radius and pitch, give

$$a_\infty = N\Gamma/(2\pi p_\infty) \tag{33}$$

and if we assume $a \approx N\Gamma/(4\pi p)$ from (4), then $p \approx p_\infty$, $a_\infty \approx 2a$. If the expanding tip vortices have constant pitch everywhere

behind the rotor, a semi-analytic determination of their influence on the flow through the rotor can be made. This is the purpose of the next Section.





## 4   The expanding Joukowsky wake with constant pitch

We assume $p$ remains constant and use the results of the previous section and the Biot-Savart law as described in the Appendix to investigate the flow immediately behind the rotor and determine the thrust and power coefficients.

When circumferentially averaged, $a$ and $v$ are determined entirely by the trailing tip vortices, whose Biot-Savart integrands in Equation (A7) are plotted in Figures 2 and 3 for $x$ close to the blade tip, in terms of axial distance $z = p\beta$ where $\beta$ is the vortex angle starting from zero at the rotor. The figures also show the small$-\beta$ asymptotes in Equation (A7) and the large$-\beta$ remainders defined in (A11) - (A13). If the tip vortex radius $t$ remains at 1, Equation (A8) gives $I_a = 1/p$ for any $x$, where $I_a$ is the Biot-Savart integral for $a$ as defined in Equation (A4), Equation (4) is recovered, and (5) remains valid. We assume that

$1/p$ is the minimum value of $I_a$, and, as explained in the Appendix, we impose $a \leq a_\infty$ so that $1/p \leq I_a \leq 2/p$. For maximum power, the familiar derivation of the Betz-Joukowsky limit suggests $R_\infty^2 \approx 2$ so we investigate $R_\infty$ around that value. Note, however, the use of Equation (4) to derive this limit means that it is applicable only to a wake that expands either very slowly, as explained above, or very rapidly to $t = \sqrt{2}$, as $I_a = 1/p$ for any constant $t$. We will show that generic wind turbine wakes at high $\lambda$ expand at a rate that is intermediate between these extremes which causes Equation (4) to be inaccurate. There is no

direct maximization of power output in the following analysis. Instead, the wake model is constrained as we now describe.

Solving (A4) for $I_a$ and $I_v$, requires $p$ and the tip vortex trajectory. We used the very simple form:

$$t = R_\infty - (R_\infty - 1)\exp(-k\beta) \tag{34}$$

which satisfies three necessary conditions: $t = 1$ when $\beta = 0$, $t \to R_\infty$ for large $\beta$, and the approach is smooth. The fourth condition is that $k$ must satisfy the reduced version of Equation (2):

$$\int_0^\infty \left(I_v^2 - I_a^2\right) x\, dx = 0. \tag{35}$$

This integral will be called the "Expansion Integral". It uniquely fixes $k$ for any choice of $R_\infty^2$ and $p$. $I_a$ and $I_v$ were obtained using the Matlab function *integral* over $\beta = [0, 200\pi]$ to an absolute tolerance of $10^{-6}$. The remainders, Equations (A12) and (A13), were then added. The expansion integral, and the mass flux integral described below, were found by trapezoidal integration using the points shown in Figure 5. The expansion integral is large for small $k$ as $v$ is (not obviously) maximized

when there is very little vortex expansion near the rotor.

The mass flux through the rotor, using (33) to remove $N\Gamma$, then determines $a_\infty$:

$$1 - a_\infty p \int_0^1 I_a x\, dx = (1 - a_\infty) R_\infty^2. \tag{36}$$

Equation (32) then yields $\lambda$. A number of possible methods were considered for solving the integral in (36). $i_a(x, \theta)$ can be written as

$$i_a(x, \theta) = \frac{d}{dx}\left(\frac{x}{d}\right) - \frac{p^2\beta^2}{d^3} \tag{37}$$

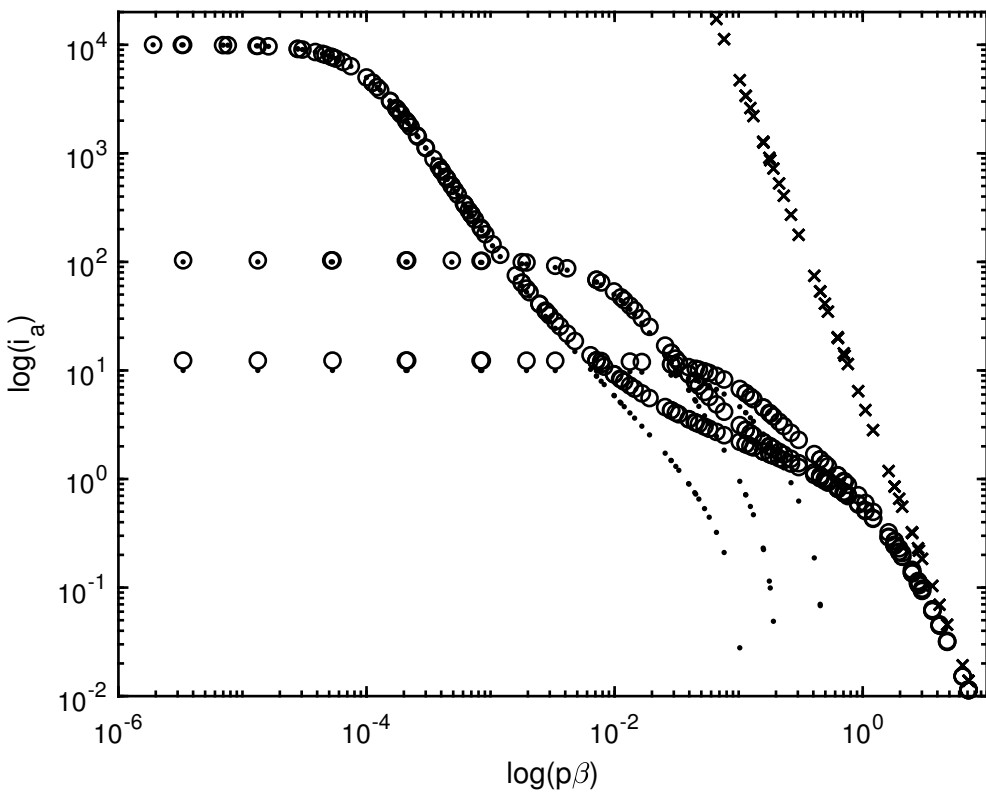

**Figure 2.** Integrand, $i_a$, for $p = 0.1, R_\infty^2 = 1.597$. Circles are $i_a$ for $x = 0.9, 0.99$, and $0.999$ from Equation (A7b). $i_a$ increases with $x$. Points shows (A10b) and $\times$ is the integrand for (A12) which is independent of $x$. For clarity, only every second data point is plotted.

which allows an analytic integration of $i_a(x, \theta)x$ in $x$. The resulting expression is complicated and probably requires numerical integration in $\theta$ and $\beta$ to obtain the mass flux. Further, the integrand is singular at a point that varies with $\theta$ and $\beta$. The simpler alternative of numerical integration of $I_a x$ was used.

To find the unique $R_\infty^2$, we impose the further condition that $k$ must match the slope of the vortex surface at the rotor. Then 265 $k$ in Equation (34) equals $k_\star$, given by

$$\frac{dt}{dz}(\beta = 0) = \frac{v(x = 1)}{1 - a(x = 1)} = (R_\infty - 1)\frac{k_\star}{p}. \tag{38}$$

The results in Table 1 were obtained using the Matlab pattern search routine *patternsearch* to minimize the single objective function that combined the magnitude of the expansion integral and $|k - k_\star|$. This, surprisingly, occurred at a constant value of $k_\star/p$, implying that the vortex surface is not dependent on $p$ or $\lambda$.

Figures 2 and 3 show the integrands $i_a$ and $i_v$ are large in the vicinity of the rotor. Their size implies that the simple assumed shape of the tip vortex trajectory, Equation (34), is reasonable, and that adding a term or terms, say, to control the approach to the far-wake would not change the analysis significantly.



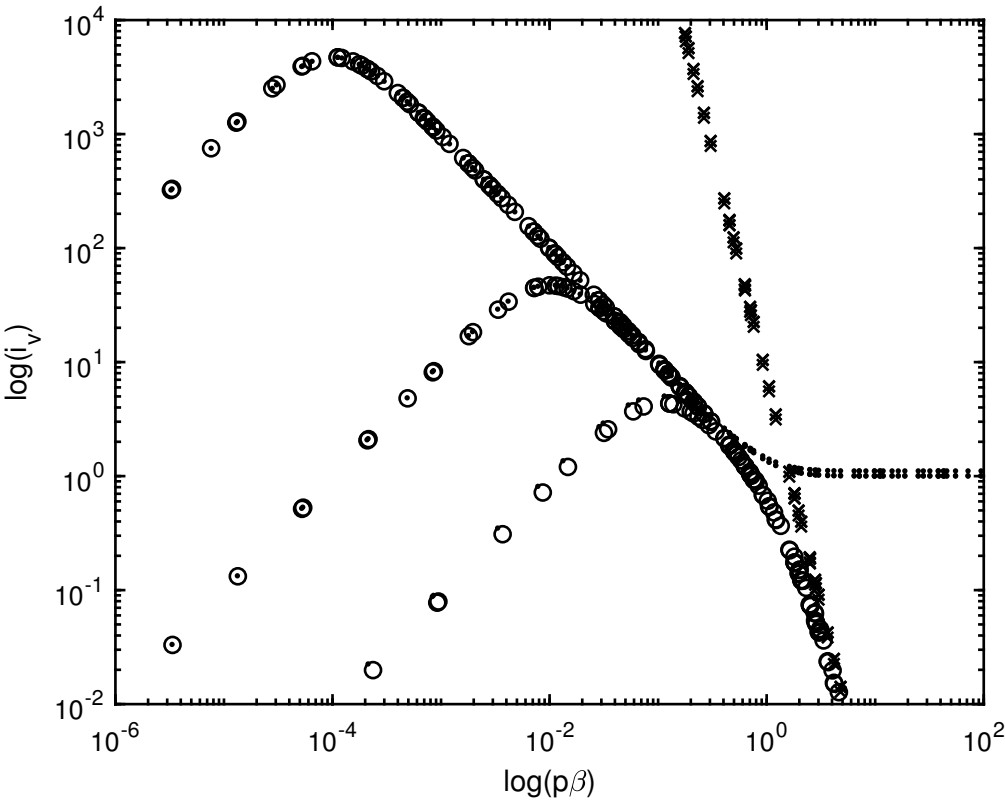

**Figure 3.** Integrand, $i_v$, for $p = 0.1, R_\infty^2 = 1.597$. Circles are $i_a$ for $x = 0.9, 0.99$, and $0.999$ from Equation (A7b). $i_a$ increases with $x$. Points shows (A10b) and $\times$ is the integrand for (A12) which is independent of $x$. For clarity, only every second data point is plotted.

| $p$ | $R_\infty^2$ | $k_*$ | $\lambda$ | $a_\infty$ | $C_P$ | $C_T$ | $C_T'$ | $\Delta C_T$ |
|------|------|------|------|------|------|------|------|------|
| 0.10 | 1.597 | 0.4947 | 7.13 | 0.574 | 0.557 | 0.819 | 0.866 | 0.087 |
| 0.05 | 1.592 | 0.2482 | 14.28 | 0.572 | 0.556 | 0.817 | 0.864 | 0.087 |

**Table 1.** Results for the expanding Joukowsky wake with constant pitch

Figure 5 shows $a$ and $v$ at the rotor for the cases in Table 1 are independent of $p$. $a(0) = 0.2956$, that is less than the Betz-Joukowsky value of 1/3, and $v(0) = 0$ as it must. The limit $a \leq a_\infty$ was applied near the blade tip, where $v$ has increased to be nearly equal, but smaller than $a$. Outside the wake, $v > a$ and takes till $x = 3$ to fall to 0.03. Similar shaped distributions of $a$ and $v$ for a Joukowsky wake are shown in Figure 5 of van Kuik (2020), who also found that Equation (2) was satisfied in his low$-\lambda$ simulations.



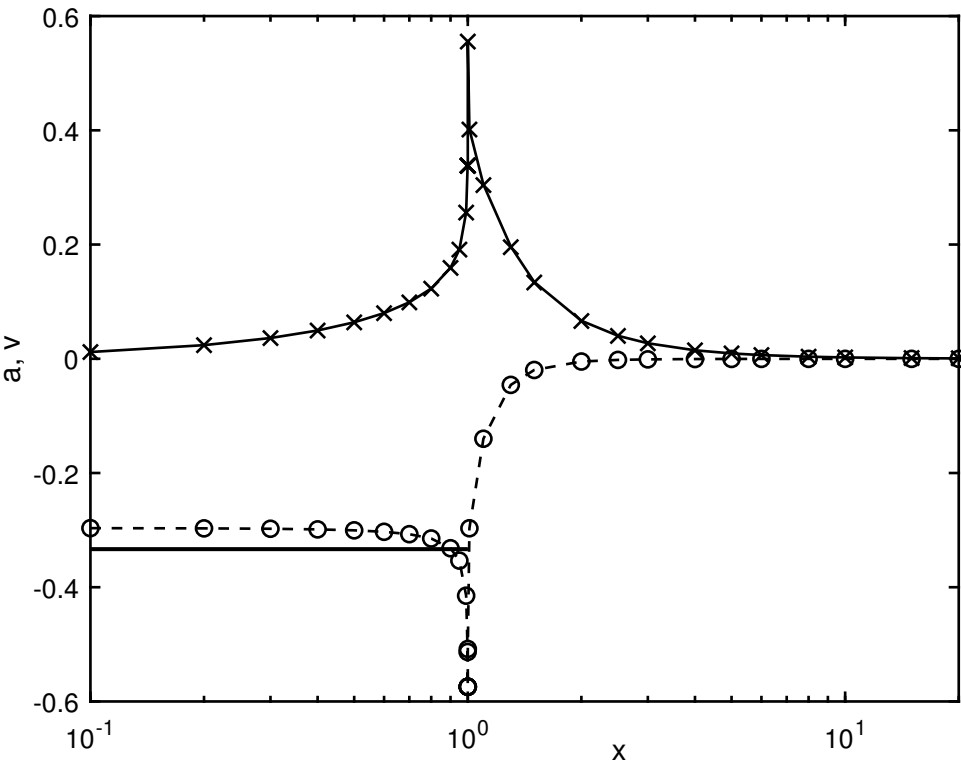

**Figure 4.** Axial induction, $a$, and radial velocity, $v$,for the conditions in Table 1. $p = 0.05 : a, \times; v, \circ. p = 0.10 : a$, dashed line; $v$, solid line. The thick solid line shows $a = 1/3$ for $x \leq 1$. Note that $-a$ is plotted for clarity, and the $x-$axis is logarithmic.

The final calculations were for $C_T$ from (1) with $w^2$ ignored because $\lambda$ is large:

$$C_T \approx N\Gamma\lambda/\pi \approx 2a_\infty p\lambda \approx 2a_\infty(1 - a_\infty/2) \tag{39}$$

using (32) and (33). We note that (1) makes the high$-\lambda$ blade element thrust constant across the rotor whereas the familiar form involving the axial velocity equation in (5) requires a significant variation near the tip. From conservation of angular momentum, and finding the power as the product of torque and angular velocity:

$$C_P \approx C_T(1 - a_\infty)R_\infty^2 \tag{40}$$

so the power extraction also decreases significantly near the tip. Equation (40) and the third component of (39) also hold for

the conventional analysis that leads to the Betz-Joukowsky limit.

Table 1 shows the biggest change from the familiar Betz-Joukowsky wake is the 20% reduction in $R_\infty^2$ which occurs because $a > a_\infty/2$ for much of the rotor, Figure 5. In other words, more of the expansion occurs upwind of the rotor. In contrast, the

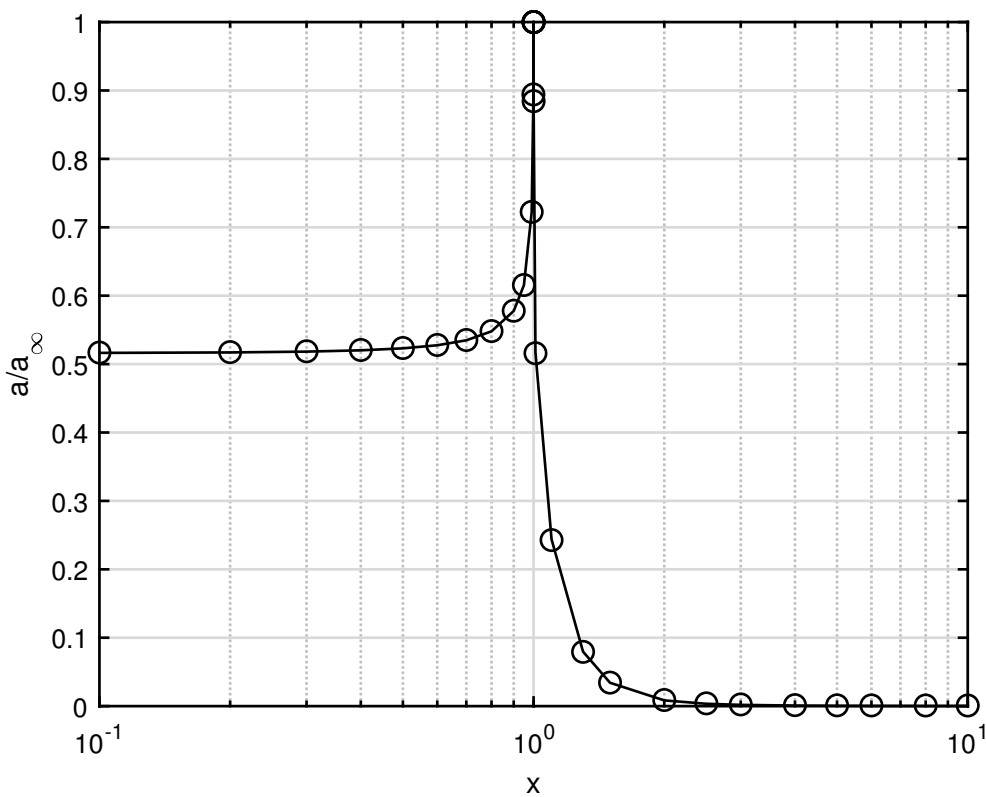

**Figure 5.** Ratio of axial induction at the rotor, $a$, to value in the far-wake, $a_\infty$. $p = 0.05 : \circ. p = 0.10$, solid line. Note that the $x-$axis is logarithmic.

maximum $C_P$ is reduced by only 6% to 0.557 and the mass flux is increased by 2%. The second last column in Table 1, gives $C'_T$, the conventional thrust coefficient evaluated from Equation (5):

$$C'_T = 8 \int_0^1 a(1-a)x dx. \tag{41}$$

$C'_T$ over-estimates $C_T$ by around 5%. Combining the last result in Equation (39) with the high$-\lambda$ limit of Equation (27) gives

$$C_T \approx C'_T + 2 \int_0^1 \delta_P x dx + 2 \int_0^1 (a^2 - v^2)x dx \tag{42}$$

where $\delta_P$ is the formal definition of the difference between $2a(1-a)$ and $-P_D/\rho$ that was considered in the last section:

$$\delta_P = 2a(1-a) + \frac{2P_D}{\rho}. \tag{43}$$



From the data in Table 1, the integral of $\delta_P$ is -0.147, so the magnitude of $P_D$ is generally significantly less than that of $P_U$. It was shown in the previous section that the pressure integrals are equal in magnitude in the minimally-expanding wake when $\lambda = 0$ but the analysis in this section shows divergence in the expanding Joukowsky wake at high $\lambda$.

## 5  Discussion

The pressure in the expanding flow ahead of a wind turbine causes a redistribution of axial momentum without altering the total thrust on the rotor. Researchers have been aware of redistributive effects for many years but the present analysis provides the first quantitative determination of them in Equations (13) and (14). Redistribution for the rotor and its constituent blade elements, depends on $a^2 - v^2$ where $v$ is the normalized radial velocity and $a$ is the usual axial induction factor. We showed that momentum is redistributed from the external flow to the turbine. Further, $v^2 - a^2$ can be used to quantify the external flow disturbed by the wind turbine and so may be useful to the study of multiple rotors in close proximity, as analyzed by, for example, Branlard and Meyer Forsting (2020) . This disturbance can be quantified by defining $I_E$ as

$$I_E = \int\limits_{x_{BS}}^{\infty} \left( v^2 - a^2 \right) x \, dx = - \int\limits_{0}^{x_{BS}} \left( v^2 - a^2 \right) x \, dx \tag{44}$$

where $x_{BS}$ is the radius of $BS$ at any $z \leq 0$. The last integral is a consequence of Equation (2) being valid for $S_U$ lying anywhere in the upwind flow. $I_E$ must be zero in the undisturbed upwind flow. It then increases to its maximum value at the rotor according to the present analysis. $I_E$ then decreases in the wake to be zero in the far-wake. In other words, the redistribution of momentum by the pressure is complete by the time the far-wake is reached. Note that the second equality in (44) does not hold in the wake.

The impulse analysis of Limacher & Wood (2020) (LW) showed that the Kutta-Joukowsky (KJ) equations for rotor thrust, Equation (1), and for the blade element contributions to the thrust, (3), are exact in the presence of wake expansion, where "exact" means using no more assumptions or approximations than the eight listed in the Introduction. The KJ equations, containing only the circumferential velocity and tip speed ratio, are not equivalent to the conventional equation involving only the axial velocity, when the flow expands. This is the outcome of the analysis in Section 4 where an expanding Joukowsky wake comprising tip and hub vortices of constant pitch was analyzed. The conventional thrust equation is altered by around 5-10%, depending on the trajectory of the tip vortices because the geometrical relation in Equation (4) is modified by the trajectory.

The first three sections of the paper used only the standard form of control volume (CV) analysis for axial momentum to determine the thrust of the rotor and the incremental thrust of the blade elements comprising the rotor. To clarify the effects of expansion, most analysis in this paper used CVs with downwind faces in the immediate vicinity of the rotor, as opposed to their common placement in the far-wake. The rotor and the flow are assumed to be circumferentially-uniform. We argued in the Introduction that the impulse analysis provides a simple and novel perspective on pressure redistribution. The thrust equations derived in Section 3 for the rotor, and in Section 4 for blade element, contain the pressure acting on the downwind face of the actuator disk, which must be removed to make the equations suitable for actual blade analysis. Removal can be done accurately only for very low tip speed ratios where the expansion and its effects, are small.





To the rotor thrust, pressure redistribution adds the integral of $a^2 - v^2$ over the rotor. This integral is equal and opposite the integral for the flow outside the wake so there is no net contribution to the thrust determined using the CVs shown in Figure 1. Unsurprisingly, the redistributive term in the blade element thrust equation also contains $a^2 - v^2$. It follows that the conventional blade element thrust equation implies $a \approx v$ but $a^2$ is generally larger than $v^2$; more precise estimates of $v$ do not appear to be possible. $a^2 > v^2$ implies that the pressure redistributes momentum from the external flow to the wake. The common derivation of the axial momentum equation which leads to the Betz-Joukowsky limit, ignores redistribution of momentum by the pressure, and then ignores the radial velocity in relating the pressure at the rear of the disk to the far-wake. These errors cancel, so the main failing of the conventional equation is the breakdown of the relation $a_\infty = 2a$ when expansion is significant. The previous Section shows this breakdown is due to the expanding tip vortices at high $\lambda$ in the Joukowsky wake. At the rotor, the slope of the streamsurface containing the tip vortices is $53°$ for maximum power extraction, Table 1, so their trajectory is intermediate between very slow or very rapid expansion, either of which would require $a_\infty = 2a$, This analysis used Equation (32) for the pitch of the tip vortex, found by moving the CV outlet to the far-wake and using LW's impulse equation for thrust.

Including $v$ in the axial momentum equation effectively adds an extra unknown to the conservation equations that may render them useless unless another equation for $u, v$ or $w$ could be derived. Further, high $v$ may cause significant alterations to the lift and drag of the blade elements near the tip. To our knowledge, radial velocity effects on airfoil lift and drag have not been studied in the context of blade element theory. We note further that van Kuik (2020) estimated the streamsurface angle at the rotor edge to be $46°$ which is close to the present value.

The role of the radial velocity and flow expansion is probably more complicated in rotors with a limited number of blades than the actuator disks considered here. Eriksen & Krogstad (2017) measured $u$, $v$, and $w$ immediately behind the rotor of a model three-bladed turbine out to a radius 20% larger than the blade tip radius. They used phase-locked averaging to obtain the flowfield as seen by an observer rotating with the blades. Significant phase variations occurred in $a$ and $v$ showing that the averages $a^2$ and $v^2$ over a blade revolution could be large even if the mean values of $a$ and $v$ are small. Nevertheless, the magnitude of both $a$ and $v$ was largest near the angular location of the blades, suggesting that the issues with radial deflection will occur in real turbines. We hope that these comments, and the present analysis, will inspire further measurements to be made far enough outside the wake to help clarify the role of flow expansion and the disturbances to the external flow.

## 6   Conclusion

Starting from the impulse-derived Kutta-Jukowsky equation for wind turbine thrust which does not involve the axial velocity, we were able to:

– Demonstrate the conventional thrust equation containing the axial velocity is correct only when the rate of wake expansion is either very small or very large, and so is accurate at low tip speed ratio.

– Derive an exact expression for the effects of flow expansion on the conventional momentum equation. This involves the axial induction factor and the radial velocity.



– Apply the conventional and impulse thrust equations in the far-wake to give the pitch of the tip vortices in the Joukowsky wake.

– Find a semi-analytic solution of the Biot-Savart law for the induced velocities at the rotor by assuming the tip vortex had constant pitch. The axial velocity near the rotor tip approached the far-wake value, but was prevented from exceeding it. The increase in the rotor value contradicts the familiar relation that the axial induction factor everywhere at the rotor is
half that of the far-wake.

– Demonstrate in Section 5 that the angle of the tip vortex surface to the wind direction was $53°$ for maximum power production, independently of the tip speed ratio and vortex pitch. Because it is neither very small nor very large, this expansion leads to an error of around 6% in the conventional thrust equation which would be accurate for both extreme expansions.

– Show the resulting wake expands less than the familiar Betz-Jukowsky wake. For two pitch values corresponding to tip speed ratios of 7 and 14, the far-wake area was unchanged at 1.59 times the rotor area.

– Find the reduction in the rotor power and thrust due to expansion. The maximum power coefficient and corresponding thrust coefficient were 6% less than the values at the Betz-Joukowsky limit.

– Quantify the influence of the expansion on the flow outside the rotor. For example, the radial velocity at three rotor radii
is still 3% of the wind speed.

*Acknowledgements.* This paper was inspired partly by an anonymous referee of LW who doubted the value of Equations (1) and (3) because they do not contain the axial velocity. We acknowledge the useful comments of Gijs van Kuik on an earlier draft of this paper. DW's contribution to this work is part of a research project on wind turbine aerodynamics funded by the NSERC Discovery Program. EL acknowledges receipt of an NSERC Post-Doctoral Scholarship.

**Appendix: The velocities behind the rotor due to an expanding Joukowsky wake of constant pitch**

This Appendix describes the use of the Biot-Savart law to determine the circumferentially-averaged velocities immediately behind the rotor. These are due entirely to the trailing vorticity: $w$ is due to the hub vortex only, whereas $u$ and $v$ result from the tip vortices only.

Without loss of generality, let the lifting line representing one blade lie instantaneously along the $x-$axis in Figure 1 and
consider the tip vortex beginning at $(1,0,0)$. We now determine the velocities induced at a point $(x,\theta,0)$ in polar co-ordinates or $(x\cos\theta, x\sin\theta, 0)$ in Cartesian co-ordinates. The vortex pitch, $p$, is assumed constant. A point on the vortex is $(t(\beta),\beta,p\beta)$ or $(t(\beta)\cos\beta, t(\beta)\sin\beta, p\beta)$ where radius $t$ is a monotonically increasing function of the vortex angle $\beta$ that asymptotes to the far-wake radius. Thus $1 \leq t \leq R_\infty$, and from here on, the dependence of $t$ on $\beta$ will be understood. An increment $dl$ along the





vortex is given by

$$dl = (-t\sin\beta + \frac{dt}{d\beta}\cos\beta, t\cos\beta + \frac{dt}{d\beta}\sin\beta, p)d\beta \tag{A1}$$

and the distance $d$ from the point to the vortex is

$$d = (x\cos\theta - t\cos\beta, x\sin\theta - t\sin\beta, -p\beta) \tag{A2}$$

so that

$$d^2 = x^2 + t^2 - 2xt\cos(\beta - \theta) + p^2\beta^2 \tag{A3}$$

which is an even function of $\beta$ and $\theta$. A straightforward application of the Biot-Savart law yields the three velocities associated with the trailing tip vortex as

$$\big(v(x,\theta), w(x,\theta), a(x,\theta)\big) = \frac{\Gamma}{4\pi}(I_v, I_w, I_a) = \frac{\Gamma}{4\pi}\int_0^\infty \frac{(i_v(x,\theta), i_w(x,\theta), i_a(x,\theta))}{d^3}d\beta \tag{A4}$$

where $\Gamma$ is the vortex strength,

$$i_v(x,\theta) = -p\left[t\beta\cos(\beta-\theta) + \left(t - \beta\frac{dt}{d\beta}\right)\sin(\beta-\theta)\right], \tag{A5a}$$

$$i_w(x,\theta) = p\left[x + \left(\beta\frac{dt}{d\beta} - t\right)\cos(\beta-\theta) - \beta t\sin(\beta-\theta)\right], \quad\text{and} \tag{A5b}$$

$$i_a(x,\theta) = t^2 - xt\cos(\beta-\theta) - x\frac{dt}{d\beta}\sin(\beta-\theta). \tag{A5c}$$

In forming the circumferential averages by integrating over $0 \leq \theta \leq 2\pi$, all the $\sin(\beta-\theta)$ terms will vanish as they are odd in $\theta$. The linearity of inviscid flow leads to equal contributions to the averaged $(u, w, a)$ from the $N$ identical and equi-spaced trailing vortices.

The simplest calculation of $i_a$ is for $x = 0$ for which the circumferential average $a(0) = a(0,\theta)$, and

$$a(0) = \frac{N\Gamma}{4\pi}\int_0^\infty \frac{t^2}{(t^2+z^2)^{3/2}}d\beta = \frac{N\Gamma}{4\pi p}\int_0^\infty \frac{t^2}{(t^2+z^2)^{3/2}}dz = \frac{N\Gamma}{4\pi p}\int_0^\infty i_a(0)dz = \frac{N\Gamma}{4\pi}I_a(0). \tag{A6}$$

$I_a$ is, clearly, dependent only on the geometry of the tip vortices. For an expanding wake with constant $p$, Equation (4) will underestimate $a$ as $I_a(x) \geq I_a(0) \geq 1/p$ when $t$ is not constant. If $p$ varied with $\beta$, then $p\beta$ in Equation(A3) would be replaced by $\int p\,d\beta$ and the direct relation between $\partial/\partial z$ and $(1/p)d/d\beta$ would be lost. It is likely that an analytic expression for the integrands in (A4) would not be possible.

Performing the $\theta-$integration of (A5) (using Mathematica) gives

$$i_v(x) = p\beta\left[(p^2\beta^2 + x^2 + t^2)/(p^2\beta^2 + (x+t)^2)\mathrm{E}(m) - \mathrm{K}(m)\right]/x/\sqrt{p^2\beta^2 + (x-t)^2}, \quad\text{and} \tag{A7a}$$

$$i_a(x) = -\left[(p^2\beta^2 + x^2 - t^2)/(p^2\beta^2 + (x+t)^2)\mathrm{E}(m) - \mathrm{K}(m)\right]/\sqrt{p^2\beta^2 + (x-t)^2}, \tag{A7b}$$





where E(.) and K(.) are the complete elliptic integrals of the second and first kind, respectively, whose argument, $m$, is given
by $m = -4xt/(p^2\beta^2 + (x-t)^2)$. Thus $v$ and $a$ can be obtained by integrating (A6) along the trajectory of the tip vortex, $t(\beta)$
for $0 \leq \beta \leq \infty$. This must, in general, be done numerically, but several checks are possible. In describing these, we continue
to use the notation $I = \int i \, d\beta$ and identify the limits to the integral if they differ from $(0, \infty)$.

If $t$ remains constant at 1, say, and the integration is over $-\infty \leq \beta \leq \infty$, that is for a doubly-infinite vortex or vortices of
constant radius and pitch, then $I_v(-\infty, \infty) = 0$ for any $x$, and

$$I_a(-\infty, \infty) = 2/p, \quad \text{for} \ \ x < 1, \tag{A8a}$$

$$= 1/p, \quad \text{for} \ \ x = 1, \ \ \text{and} \tag{A8b}$$

$$= 0, \quad \text{otherwise.} \tag{A8c}$$

The interior and exterior solutions, Equations (A8a) and (A8c), are consequences of the Kawada-Hardin (KH) equations,
Kawada (1936), Hardin (1982), derived from the velocity potential. All equations in (A8) follow from (A5a) and (A5c). Using
*NIntegrate* in Mathematica and Matlab's *integral*, these results were reproduced to six significant figures for a similar range of
$x$ to that used in the main text and limits of $\pm 1000\pi$ on the integration. For a singly-infinite helix, the values of $I_a$ for $z = 0$
when $\beta = 0$, are half those in (A8). These were reproduced numerically to the same accuracy. $I_v$ is not available from the KH
equations for this case.

As with any Biot-Savart analysis, the behaviour of Equation (A7) as $x \to t(0) = 1$ must be considered. This was done by
first transforming the arguments of the elliptic integrals to positive values using

$$K(-m) = K(m_p)/\sqrt{m+1} \ \ \text{and} \ \ E(-m) = \sqrt{m+1} \, E(m_p) \tag{A9}$$

where $m_p = m/(m+1)$. The transformations are easily derived from the definition of the integrals. As $m_p \to 1$, $E(m_p) \sim 1$,
Formula (19.6.1) of DLMF (2021), and $K(m_p) \sim \log(16/m_p')/2$ where $m_p' + m_p = 1$, Formula (17.3.26) of Abramowitz &
Stegun (1964). The leading terms in (A7) become

$$i_v(x) \sim p\beta(p^2\beta^2 + x^2 + t^2)/\sqrt{p^2\beta^2 + (x+t)^2}/x/\left(p^2\beta^2 + (x-t)^2\right), \ \ \text{and} \tag{A10a}$$

$$i_a(x) \sim -(p^2\beta^2 + x^2 - t^2)/\sqrt{p^2\beta^2 + (x+t)^2}/\left(p^2\beta^2 + (x-t)^2\right). \tag{A10b}$$

showing that a logarithmic singularity occurs in $i_a$ despite it being the integrand for the circumferentially-averaged axial
velocity. This is a stronger singularity than that in Chattot's (2020) perturbation analysis of the flow near the edge of the rotor,
which assumes a vortex cylinder wake. There is no logarithmic singularity in $i_v$, but the slope $di_v/d\beta$ increases without bound
as $\beta \to 0$. These behaviours could be mitigated by using the well-known "cut-off" modification to the limits of the Biot-Savart
integrals as was done for helical vortices by Ricca (1994), see also Section 11.2 of Saffman (1992). There is, however, a
simpler, heuristic alternative. The upper limit on $a(x)$ as $x \to 1$ is taken to be $a_\infty$ as a consequence of assuming $p$ is constant in
the wake. Thus $I_a \leq 2/p$ was enforced in the calculations described in the main text. Whenever this was done, $I_v$ was assumed
equal to the maximum value below the limit on $I_a$.





The numerical evaluation of $I_a$ and $I_v$ can be improved by considering the asymptotic behaviour of $i_a$ and $i_v$ for large $\beta$. The leading terms are simple functions of $\beta$, allowing the infinite integrals to be approximated as

$$I_a(x) \approx I_a(x, \widehat{\beta}) + R_a(\widehat{\beta}) \tag{A11}$$

where the first term was obtained numerically over $\beta = [0, \widehat{\beta}]$ and the remainder, $R_a(\widehat{\beta})$, is an approximation to the integral over $\beta = [\widehat{\beta}, \infty]$. $R_a(\widehat{\beta})$ is independent of $x$:

$$R_a(\widehat{\beta}) = \pi R_\infty^2/(2\widehat{\beta}^2 p^3). \tag{A12}$$

This result also follows from (A6) when $z >> R_\infty$. For $I_v$, the remainder varies across the rotor:

$$R_v(x, \widehat{\beta}) = \pi R_\infty^2 x/(2\widehat{\beta}^3 p^4). \tag{A13}$$

It was found that $\widehat{\beta} = 200\pi$ was sufficient to ensure six-figure accuracy of the integrals over the range of $x$ considered in the main text. By this method, $I_v$ converged faster than $I_a$, reaching 99% of the final value by $\beta = 2\pi$ for any $x$.



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
