# Peer review of "Some effects of flow expansion on the aerodynamics of horizontal-axis wind turbines"

_Wind Energy Science, 2021_

## Referee Comment (RC1)

Review of WES-2021-52:

**Some effects of flow expansion on the aerodynamics of horizontal axis wind turbines,**

by David H. Wood and Eric J. Limacher

Reviewer: Gijs van Kuik, TU-Delft

Introduction

Performance calculation of wind turbines is based on momentum theories developed over the last 100 years, with the turbine's rotor replaced by a disc. Most easy is the calculation for the entire rotor (disc) with a high tip speed ratio of the rotating force field, giving disc-averaged results. The calculation methods for low tip speed ratio and for the distribution of velocities along the radius of the disc are less developed. The paper addresses both areas, so it is very much welcomed.

As far as I have checked there are no mathematical flaws. The reference list shows a good knowledge of the relevant literature. Being a theoretical paper, it is not easy to read. The has a logical structure, but at several places accuracy of the text could be better, and the explanation in physical terms could be expanded (below I have addressed this).

The reviewer

The authors refer several times to van Kuik (2018) and van Kuik (2020), so to my own work. This is one of the reasons to act as a non-anonymous reviewer: in this way authors and readers know that the reviewer discusses the results of the paper with his own work in mind. At some places in the paper I conclude that the results of the authors and reviewer differ. The intention of these remarks is not to say that the authors' results are wrong (it might turn out that my results are wrong), but that an explanation of the differences is required.

General remarks

1- Disc or rotor or turbine: all are used, but sometimes in a confusing way. E.g. line 30 : 'a circumferentially uniform rotor'. I advise to use the word disc for this, as a rotor is characterized by a finite number of blades. When this number increases to infinity, or when 'circumferentially uniform' is used, it is a disc. The word turbine in the first sentence of the Discussion is not well in place. I propose to change the title of section 3, as there are no blade elements. The words 'blade element' has the annotation of an aerofoil with a finite length chord, with lift and drag acting on it. None is present here, so better would be: Local thrust in expanding flow. This improves the structure of the paper: section 2 treats the entire disc and stream tube, section 3 the local properties or the distribution along the chord. It took me a while to recognize this structure using the titles as they are. A suggestion for section 4 is: start a new section Results at line 278.
2- At several places the authors claim that a topic or result was lacking in the literature, or presented by them for the first time. Please be extremely careful with this. In the detailed comments presented below I give some examples where the claim is not entirely justified.
3- The list of assumptions at page 3: This is very much appreciated, as in general we do not like assumptions so it is good to list these explicitly. However, in the remainder of the paper more assumptions are being made which worry me much more than the ones in the

list. These extra assumptions allow the derivation of the results and a numerical assessment but maybe at the cost of the results:

1- the pitch of the convected vorticity is constant even in the expanding part of the flow,
2- the pressure gradient due to the swirl results in undisturbed pressure at the boundary of the far wake, and
3- the pressure jump across the disc is anti-symmetric: $P_U$ is assumed to be - $P_D$.

Please list these assumptions as explicitly as those at page 3, at as convenient place. These assumptions may be the cause of the remarkable results in Table 1. Although the calculated distribution of the velocity is very familiar to results in literature, the performance results in Table 1 are not. For lambda = 14, the effect of swirl in the wake is known to be very minor, so the result for the disc with infinite lambda should be recovered, which isn't. This is something to elaborate on. Which assumption has which effect on these results?

4- The authors use a nice result of their preceding paper, eq. (2), to quantify what is called the redistribution of momentum. This is the result of the pressure acting from one streamline to another, resulting an axial contribution to the local momentum balance. I fully agree with this, but not with the conclusion that 'momentum is redistributed from the external flow to the turbine'. The external flow is the flow not passing the actuator disc, so the stream tube boundary splits the flow in external and disc-passing flow. As known from literature and confirmed by the authors, the contribution of the pressure - acting on this boundary - to the momentum balance of the flow passing the disc, is 0. There is no momentum redistribution from the external flow to the stream tube, irrespective of the tip speed ratio.

5- It is appreciated when an explanation in physical terms is given of what redistribution is, where it takes place. The authors conclude that the redistribution is 'complete by the time the far wake is reached'. This is a consequence of the fact that the pressure only can contribute when the flow is expanding, partly upwind of the disc, partly downwind. At several places the authors emphasize that it is the upwind part of the flow responsible for the redistribution. Given the choice of the control volumes in figure 1 the upwind part (including the disc) is used to derive results, but physically there is no reason to exclude the downwind part (see also detailed remarks below).

Detailed remarks

Line 27: the connection of the sentence starting with 'This ..' in connection with the sentence in lines 26-26 is not clear to me. In CV1 and CV2 there is no axial component of the pressure acting on the boundary with constant $R_{cv}$.

Line 28-30: About the distribution of momentum by pressure forces acting on the control volume: The word 'redistribution' is new as far as I know, and well in place. I disagree with 'it is lacking'. It has been treated and quantified in vanKuik(2018), section 5.2.4 & figure 5.13 for Froude discs (infinite tip speed ratio) and section 6.4.5 & figure 6.13 for Joukowsky discs (finite tip speed ratio). The momentum balance per annulus contains the contribution by the pressure at the boundary of the annulus. The pressure contribution was numerically calculated.

Line 39: 'was assumed', why not 'is assumed'? You use this property in line 72.

Page 3, Assumptions:

> 1: here are two assumptions: about the flow and about the rotor (disc?)
>
> 3&5: as 1 gives inviscid flow, these are superfluous
>
> 4: I agree, but it follows from the continuity equation for incompressible flow
>
> 6: follows by the assumption that the flow far upwind of the rotor is uniform
>
> 8: as vorticity is absent upwind of SU, this applies to the space between SU and SD? Am I right? About blade element: see general remark 1
>
> Missing (?) assumption: I assume that you apply the Joukowsky model for the wake throughout the paper, as indicated in line 218. If so, it is worth to mention in the list.

Line 82: lambda = 0: this flow case has been studied by van Kuik (2018), section 6.3.1 and figure 6.2. The flow is blocked (axial velocity at the disc and in the wake is 0, so a = 1), and has a significant wake expansion. The wake has swirl with w = 1.648 R/r, so $w^2$ is not 4a. Please compare your solution with this.

Line 95-98: as for eq. (4) and (5) and section 4, the result mentioned in these lines depends on the pitch being constant. I have three remarks: 1: This restriction is not used by van Kuik (2018), section 6, still giving the same result that $a_\infty \approx 2a$ for infinite lambda, and < 2a for finite lambda. Please discuss the impact of constant pitch p in section Discussion.  2: Equation 4 is taken from Okulov&Sorensen (2008), but it holds for the Betz/Goldstein distribution of w, not the Joukowsky distribution applied here. Still this may be justified (and you are in good company as Prandtl's tip correction is based on the same crossover) but then this should be told. 3: see general remark 3.

Line 114, eq. (11) this equation is a crucial equation in the reasoning of the paper. When the first and third integral are combined and evaluated, they become the integral of PU, as it should be. As eq. 11 is valid for the entire disc, redistribution is not that clear to me as no information is involved regarding redistribution over the disc surface. The fact that (a2-v2) appears is not the same as redistribution, unless my understanding of redistribution is not correct.

Line 124, equation 13 holds for the upwind half of the stream tube. Using eq. (2) the right hand side may be converted to an integral from 1 to infinity, preceded by a minus sign. Using a momentum balance for the external flow combined with Bernoulli's equation, this modified eq. (13) is valid for all z, including downwind. My interpretation is that at any position z the axial contribution of the pressure at BS, integrated from upwind to z, equals the modified right hand side at that position z. Far enough downwind the right hand sides becomes 0, so the modified eq. (13) is a new proof that the pressure at the BS does not contribute. Do you agree?

Line 126: You write 'It is easy to generalize (13)' but is it that easy? Some intermediate steps would help to show that the Bounding Streamline in (13) can be replaced by any streamline leading to (14). If this interpretation is correct, line 128 is confusing, as BS belongs to (13) not (14).

Line 132-133: I do not see the need to use the unsteady Bernoulli equation. (16) follows by applying steady Bernoulli at a stream surface from the downstream disc side to the far wake.

Line 140, eq. 18: this holds for a Joukowsky wake, with a root vortex at the axis controlling w, see my remark on: page 3 Assumptions.

Line 147: this is a severe assumption, see general remark 3. This assumption limits the applicability to high tip speed ratios. In van Kuik (2018), section 6.2.3, it is shown that an additional pressure term is required to have the pressure undisturbed at the wake boundary. It is this term (combined with the expansion ratio $R\infty/R$) that determines the deviation from the classical result $a = 0.5 \, a_\infty$ in case of low tip speed ratios, see section 6.2.5 of van Kuik (2018).

Line 153: see my remark at line 82.

Line 174: see remark at line 28-30

Line 180 - 185: if the redistribution term ($a^2$-$v^2$) expresses the pressure exerted by one annulus to another, it contributes only to an axial momentum when there is flow expansion. Indeed the process is done when the wake becomes the far wake. I am confused by the sentence about the redistribution probably being complete everywhere within the wake. What is 'within the wake'? In radial direction? In axial direction? A combination?

Line 197: As remarked before, I disagree. There is no redistribution from external to internal flow,. There is momentum transfer from one streamline to another. inside the stream tube. If that is what you mean I agree, but like to see the text changed accordingly.

Line 198: what are stationary rotors?

Line 199: remove 'because flow expansion is neglected'

Line 205-210: the relevance of this paragraph is not that clear, and the text is not that clear. Which first term? About lambda = 0: vanKuik(2018) shows that circulation Gamma is independent of x, section 6.3.1. Suddenly the number of blades plays a role. If N becomes important, there is no circumferential uniformity any more, as in the analyses so far. Is this something to consider?

Line 233: see my remark at lines 95-98.

Now I jump to:

Table 1: is the last column correct? Delta Ct is not explained but I assume it is the difference between the 7$^{th}$ and 8$^{th}$ column. If so, there is an error in the table. Even for lambda = 14.28 the deviation from the Betz-Joukowsky limit is significant. Although the plots in figures 4 and 5 show the same characteristics as those in van Kuik(2018) the performance results differ: for lambda = 5 the difference with the Betz-Joukowsky limit was <1%, decreasing with higher lambda. See my next remark

Line 298 The discussion is partly a summary of results. I would prefer a discussion about the consequences of the assumptions made, and how these are impacting the results in Table 1 (see also General remarks)

Line 301-302 It may be a matter of language, but this statement is too strong. Eqs (12,13,14) are new indeed, but they may be considered as another way of expressing the thrust. Methods

used so far do not use $P_D$ or $(a^2 - v^2)$ but express T in terms of the rotational speed or lambda and the circulation or swirl around the axis, see eq. 4.6 in Sørensen(2016) and 4.23 in vanKuik(2018).

Line 332/333: this statement is not correct. The 'common derivation' of the B-J performance is the derivation for infinite lambda. It does not need information about the distribution of the axial and radial velocity at the disc, nor does it predict the distribution. So what you use 'ignores' I would say 'does not need'. Things are different when you consider annuli instead of the entire stream-tube. Indeed no closed form solution is available, but an analysis (with calculated pressure) is as mentioned in my remark on lines 28-30.

Finally: Abstract and Conclusions : I expect that the abstract and the conclusions will be modified when (some of) my remarks have been taking into account.

---

## Referee Comment (RC4)

[referee-annotated manuscript omitted]

---

## Author Comment (AC1)

Review of WES-2021-52:

**Some effects of flow expansion on the aerodynamics of horizontal axis wind turbines, by David H. Wood and Eric J. Limacher**

Reviewer: Gijs van Kuik, TU-Delft

Introduction

Performance calculation of wind turbines is based on momentum theories developed over the last 100 years, with the turbine's rotor replaced by a disc. Most easy is the calculation for the entire rotor (disc) with a high tip speed ratio of the rotating force field, giving disc-averaged results. The calculation methods for low tip speed ratio and for the distribution of velocities along the radius of the disc are less developed. The paper addresses both areas, so it is very much welcomed.

As far as I have checked there are no mathematical flaws. The reference list shows a good knowledge of the relevant literature. Being a theoretical paper, it is not easy to read. The has a logical structure, but at several places accuracy of the text could be better, and the explanation in physical terms could be expanded (below I have addressed this).

The previous discussion of the relationship between the Kutta-Joukowsky thrust equation (involving w) and the conventional thrust equation (involving a) has been modified at the end of Section 1.   We show the latter is invalid at $\lambda = 0$ because the thrust is balanced by the pressure acting on the rear face of the disc ($P_D$ in our notation).  The conventional equation is shown the be approximately valid at higher $\lambda$. Minor errors have been corrected in the new Equations (49) and (50) and all numerical results have been recalculated, figures 2-5 inclusive have been redrawn, and the entries in table 1 checked.  No significant changes were found. We have taken the opportunity to rewrite the new equations for the Biot-Savart integrands in the more usual form of (43) and (44) and to move the material in the appendix to Section 4 for better continuity of presentation.

The reviewer

The authors refer several times to van Kuik (2018) and van Kuik (2020), so to my own work. This is one of the reasons to act as a non-anonymous reviewer: in this way authors and readers know that the reviewer discusses the results of the paper with his own work in mind. At some places in the paper I conclude that the results of the authors and reviewer differ. The intention of these remarks is not to say that the authors' results are wrong (it might turn out that my results are wrong), but that an explanation of the differences is required.

We appreciate the detailed and comprehensive comments by Dr van Kuik.  Our responses are given in red below each comment and amended text is shown in blue in the revised manuscript. We acknowledge some differences in interpretation and expand on ours in the responses.

General remarks

1- Disc or rotor or turbine: all are used, but sometimes in a confusing way. E.g. line 30 : 'a circumferentially uniform rotor'. I advise to use the word disc for this, as a rotor is

characterized by a finite number of blades. When this number increases to infinity, or when 'circumferentially uniform' is used, it is a disc. The word turbine in the first sentence of the Discussion is not well in place. I propose to change the title of section 3, as there are no blade elements. The words 'blade element' has the annotation of an aerofoil with a finite length chord, with lift and drag acting on it. None is present here, so better would be: Local thrust in expanding flow. This improves the structure of the paper: section 2 treats the entire disc and stream tube, section 3 the local properties or the distribution along the chord. It took me a while to recognize this structure using the titles as they are. A suggestion for section 4 is: start a new section Results at line 278.

The term "disc" has been added to the abstract and the text has been revised to clarify when the approximation is made. We have not changed the word "turbine" at the start of the Discussion because its use is valid in context: the effect of pressure on the bounding stream surface occurs for any number of blades whereas our equations (14) and (15) are strictly valid only for a disc.  We have moved the material from the Appendix to Section 4 to improve the flow of the analysis, and added two subsections along the lines of the suggestion.

2- At several places the authors claim that a topic or result was lacking in the literature, or presented by them for the first time. Please be extremely careful with this. In the detailed comments presented below I give some examples where the claim is not entirely justified.

We respond to the specific comments below.

3- The list of assumptions at page 3: This is very much appreciated, as in general we do not like assumptions so it is good to list these explicitly. However, in the remainder of the paper more assumptions are being made which worry me much more than the ones in the list. These extra assumptions allow the derivation of the results and a numerical assessment but maybe at the cost of the results:

1- the pitch of the convected vorticity is constant even in the expanding part of the flow,
2- the pressure gradient due to the swirl results in undisturbed pressure at the boundary of the far wake, and
3- the pressure jump across the disc is anti-symmetric: $P_U$ is assumed to be - $P_D$.  Please list these assumptions as explicitly as those at page 3, at as convenient place. These assumptions may be the cause of the remarkable results in Table 1. Although the calculated distribution of the velocity is very familiar to results in literature, the performance results in Table 1 are not. For lambda = 14, the effect of swirl in the wake is known to be very minor, so the result for the disc with infinite lambda should be recovered, which isn't. This is something to elaborate on. Which assumption has which effect on these results?

The 8 assumptions listed in the Introduction apply to the derivation of Equations (1) and (2). The main reason to give them is to show those equations are valid in the presence of flow expansion, in contrast to the conventional axial momentum equation on the right side of (6), and that the resulting thrust equations (1-3) are correct for any distribution of w (or circulation).  Concerning the subsequent assumptions:

1. The extra text at the end of Section 3 provides part of the justification for assuming constant pitch: we want to extend the KH equations to expanding helical vortices and assuming constant pitch allows this to be done analytically up to the final integration in β. We also point out that constant pitch is also implied in the conventional relation a = 2a∝. Thus the departures we find from that equation are due to vortex expansion.

2. This is discussed on L150ff. We believe the original text is sufficient as it points out that any difference in pressure at the boundary of the far-wake is likely to be small at large λ.

3. There are no additional assumptions made in deriving the pressure jump across the disc; rather, we have drawn conclusions about $P_U$ and $P_D$. We have amended the text with the intention of improving the clarity. In summary: the jump is asymmetric for all operating λ.

4- The authors use a nice result of their preceding paper, eq. (2), to quantify what is called the redistribution of momentum. This is the result of the pressure acting from one streamline to another, resulting an axial contribution to the local momentum balance. I fully agree with this, but not with the conclusion that 'momentum is redistributed from the external flow to the turbine'. The external flow is the flow not passing the actuator disc, so the stream tube boundary splits the flow in external and disc-passing flow. As known from literature and confirmed by the authors, the contribution of the pressure acting on this boundary - to the momentum balance of the flow passing the disc, is 0. There is no momentum redistribution from the external flow to the stream tube, irrespective of the tip speed ratio.

The reviewer is correct. We intended to use "redistribution" as a single term describing the effect of non-zero pressure in the expanding wake but concede that it was not a good choice. All references to "redistribution" have been removed.

5- It is appreciated when an explanation in physical terms is given of what redistribution is, where it takes place. The authors conclude that the redistribution is 'complete by the time the far wake is reached'. This is a consequence of the fact that the pressure only can contribute when the flow is expanding, partly upwind of the disc, partly downwind. At several places the authors emphasize that it is the upwind part of the flow responsible for the redistribution. Given the choice of the control volumes in figure 1 the upwind part (including the disc) is used to derive results, but physically there is no reason to exclude the downwind part (see also detailed remarks below).

The situation behind the rotor is discussed in L185ff where we have modified the text to make clearer the meaning. It is more complicated than in the upwind flow and, fortunately, the important results for the actuator disc (14) and (15) can be derived without considering the wake.

Detailed remarks

Line 27: the connection of the sentence starting with 'This ..' in connection with the sentence in lines 26-26 is not clear to me. In CV1 and CV2 there is no axial component of the pressure acting on the boundary with constant Rcv .

We have modified the text starting at L25 to hopefully improve the clarity.

Line 28-30: About the distribution of momentum by pressure forces acting on the control volume: The word 'redistribution' is new as far as I know, and well in place. I disagree with 'it is lacking'. It has been treated and quantified in vanKuik(2018), section 5.2.4 & figure 5.13 for Froude discs (infinite tip speed ratio) and section 6.4.5 & figure 6.13 for Joukowsky discs (finite tip speed ratio). The momentum balance per annulus contains the contribution by the pressure at the boundary of the annulus. The pressure contribution was numerically calculated.

We stand by our statement that a general closed form expression of the pressure contribution "is lacking". We have referred to van Kuik (2018) as an example of prior treatment.

Line 39: 'was assumed', why not 'is assumed'? You use this property in line 72.

We were adhering to the convention that prior work is referred to in the past tense. We have added a sentence to state that the assumption is used in the present work.

Page 3, Assumptions:

     1: here are two assumptions: about the flow and about the rotor (disc?)

The text has been altered.

     3&5: as 1 gives inviscid flow, these are superfluous

Assumption 1 covers the upwind and external flow only. #3 comes from the form of the stress tensor on the control volume boundaries in Noca's (1997) derivation of the impulse equations – see LW for details and the reference. #5 is a separate assumption because, for example, the 2D version of the thrust analysis of LW does not reduce to the conventional equation for the drag on a non-lifting 2D body because it ignores the radial vorticity (in the present context, the spanwise vorticity in 2D flow) that carries that drag.

     4: I agree, but it follows from the continuity equation for incompressible flow

     6: follows by the assumption that the flow far upwind of the rotor is uniform

These two comments are correct: the assumptions are listed for completeness.

     8: as vorticity is absent upwind of SU, this applies to the space between SU and SD? Am I right? About blade element: see general remark 1

The text has been altered.

     Missing (?) assumption: I assume that you apply the Joukowsky model for the wake throughout the paper, as indicated in line 218. If so, it is worth to mention in the list.

There are no additional assumptions about the wake structure. In other words, Equations (1-3) are valid for any w(x). This covers the range from $\lambda = 0$ where the optimum circulation is quadratic in x, to high $\lambda$ where we expect a Joukowsky wake to apply with circulation inversely proportional to x.

Line 82: lambda = 0: this flow case has been studied by van Kuik (2018), section 6.3.1 and figure 6.2. The flow is blocked (axial velocity at the disc and in the wake is 0, so a = 1), and

has a significant wake expansion. The wake has swirl with w = 1.648 R/r, so $w^2$ is not 4a. Please compare your solution with this.

The statement in red above about the quadratic dependence of circulation for a stationary rotor comes from the lifting line analysis of Wood, D. H. (2015). Maximum wind turbine performance at low tip speed ratio. Journal of Renewable and Sustainable Energy, 7(5), 053126, which derives the optimum low λ performance: w ~ x when λ = 0, and a is negligible; our previous result $w^2$= 4a was incorrect.  In a separate study, we are working on improving the lifting line analysis for low λ, and have reproduced that result, but have not yet submitted the work for publication. The relationship between w and a can be viewed in another way: Equation (3) is valid at any λ and any distribution of w(x).  The manuscript has been modified to show that the conventional momentum equation in terms of a is invalid at λ = 0 but can be approximately correct at higher λ.  We are currently studying this important issue as part of the work mentioned above. We clearly have a different view on low-λ flows from Dr van Kuik but the issue needs further analysis to be resolved.

Line 95-98: as for eq. (4) and (5) and section 4, the result mentioned in these lines depends on the pitch being constant. I have three remarks: 1: This restriction is not used by van Kuik (2018), section 6, still giving the same result that $a_\infty \approx 2a$ for infinite lambda, and < 2a for finite lambda. Please discuss the impact of constant pitch p in section Discussion.   2: Equation 4 is taken from Okulov&Sorensen (2008), but it holds for the Betz/Goldstein distribution of w, not the Joukowsky distribution applied here. Still this may be justified (and you are in good company as Prandtl's tip correction is based on the same crossover) but then this should be told. 3: see general remark 3.

We emphasize that we do not assume the Joukowsky wake model until Section 4 where its use is limited to high λ. The geometric Equation (4) is valid for any constant pitch, constant radius singly-infinite helicoidal vortex and any distribution of Γ.  As mentioned above, we have expanded the discussion of the reasons for assuming constant pitch in Section 4.  The results show that a approaches $a_\infty$ near the blade tip at high expansion, which does not occur for trailing vortices of constant pitch and radius.  The result $a \rightarrow a_\infty$ is then attributed to vortex expansion. We have added a reference to the experiments of Krogstad & Adamarola (2012) who found $a \approx 0.8$ near the edge of a three-bladed model turbine at high λ. Unfortunately they did not measure the far-wake but it is clear that $a \rightarrow a_\infty$ in this case.

Line 114, eq. (11) this equation is a crucial equation in the reasoning of the paper. When the first and third integral are combined and evaluated, they become the integral of PU, as it should be. As eq. 11 is valid for the entire disc, redistribution is not that clear to me as no information is involved regarding redistribution over the disc surface. The fact that (a2-v2) appears is not the same as redistribution, unless my understanding of redistribution is not correct.

As mentioned above, we have removed all references to redistribution from the text.

Line 124,  equation 13 holds for the upwind half of the stream tube. Using eq. (2) the right hand side may be converted to an integral from 1 to infinity, preceded by a minus sign. Using a momentum balance for the external flow combined with Bernoulli's equation, this modified eq. (13) is valid for all z, including downwind. My interpretation is that at any position z the

axial contribution of the pressure at BS, integrated from upwind to z, equals the modified right hand side at that position z. Far enough downwind the right hand sides becomes 0, so the modified eq. (13) is a new proof that the pressure at the BS does not contribute. Do you agree?

The purpose of what is now Equation (15) was simply to point out that the term involving $a^2-v^2$ describes the axial force due to pressure anywhere in the upwind flow. Since the arguments about the relative magnitude of a and v at the disc in the text, show that the right side of Equation (14) is positive in general, then (13) shows that it contributes to the rotor thrust.

Line 126: You write 'It is easy to generalize (13)' but is it that easy? Some intermediate steps would help to show that the Bounding Streamline in (13) can be replaced by any streamline leading to (14). If this interpretation is correct, line 128 is confusing, as BS belongs to (13) not (14).

We think the answer is yes, because what are now (14) and (15) must hold for any axisymmetric, expanding inviscid flow with or without a rotor. The reference to BS was intended only to link the nomenclature.

Line 132-133: I do not see the need to use the unsteady Bernoulli equation. (16) follows by applying steady Bernoulli at a stream surface from the downstream disc side to the far wake.

The unsteady form is needed as explained in the appendix B to LW and is needed for the general case of finite N.

Line 140, eq. 18: this holds for a Joukowsky wake, with a root vortex at the axis controlling w, see my remark on: page 3  Assumptions.

What is now Equation (19) must be true in the far-wake at sufficiently high λ for any distribution of w.  It is a consequence of the Navier-Stokes equations in cylindrical polar co-ordinates with axial and circumferential uniformity, with a similar assumption to #3 made about the viscous or Reynolds stress terms.

Line 147: this is a severe assumption, see general remark 3. This assumption limits the applicability to high tip speed ratios. In van Kuik (2018), section 6.2.3, it is shown that an additional pressure term is required to have the pressure undisturbed at the wake boundary. It is this term (combined with the expansion ratio R∞/R) that determines the deviation from the classical result $a = 0.5\, a_\infty$ in case of low tip speed ratios, see section 6.2.5 of van Kuik (2018).

The context of the assumption is high λ, so it is not "severe". We have shown a deviation from $a = 0.5 a_\infty$ at high λ without any significant pressure effects.  Whether they are important at low expansion (low λ) is a separate matter.

Line 153: see my remark at line 82.

We don't have anything to add.

Line 174: see remark at line 28-30

"redistribution" and similar terms have been removed.

Line 180 - 185: if the redistribution term ($a^2$-$v^2$) expresses the pressure exerted by one annulus to another, it contributes only to an axial momentum when there is flow expansion. Indeed the process is done when the wake becomes the far wake. I am confused by the sentence about the redistribution probably being complete everywhere within the wake. What is 'within the wake'? In radial direction? In axial direction? A combination?

The statement about completeness has been removed.

Line 197: As remarked before, I disagree. There is no redistribution from external to internal flow,. There is momentum transfer from one streamline to another. inside the stream tube. If that is what you mean I agree, but like to see the text changed accordingly.

In removing "redistribution" we have introduced "interchange" at several locations. This is our version of "momentum transfer"

Line 198: what are stationary rotors?

$\lambda = 0$. This has been added.

Line 199: remove 'because flow expansion is neglected'

We have left the statement in place because there must be some expansion even for stationary rotors.

Line 205-210: the relevance of this paragraph is not that clear, and the text is not that clear. Which first term? About lambda = 0: vanKuik(2018) shows that circulation Gamma is independent of x, section 6.3.1. Suddenly the number of blades plays a role. If N becomes important, there is no circumferential uniformity any more, as in the analyses so far. Is this something to consider?

As mentioned above, the lifting line analysis of Wood (2015) shows the optimum loading of stationary rotors is linear in radius, not independent of x. The role of N is interesting because the non-uniformity associated with finite N, comes through the $w^2$ term in the exact KJ thrust equation and this term becomes negligible at high thrust. Thus we claim the statements referred to are valid for all N at high $\lambda$.

Line 233: see my remark at lines 95-98.

We do not have anything to add to our response.

Now I jump to:

Table 1: is the last column correct? Delta Ct is not explained but I assume it is the difference between the 7th and 8th column. If so, there is an error in the table. Even for lambda = 14.28 the deviation from the Betz-Joukowsky limit is significant. Although the plots in figures 4 and 5 show the same characteristics as those in van Kuik(2018) the performance results differ: for lambda = 5 the difference with the Betz-Joukowsky limit was <1%, decreasing with higher lambda. See my next remark

The error in the Table has been corrected and the discussion of the pressure asymmetry across the disc has been simplified by using the new Equation (59). We have modified the

conclusions to emphasize that the results on optimum power, wake expansion etc are results of our model with its specific assumptions.

Line 298 The discussion is partly a summary of results. I would prefer a discussion about the consequences of the assumptions made, and how these are impacting the results in Table 1 (see also General remarks)

We have modified the text to emphasize that the most important assumption is that the tip vortices expand in a manner that is consistent with the expansion integral and the matching of vortex and stream surfaces. All the important results in Table 1 follow from that. We have also discussed the possible consequences of our analysis for future work. As such, some summary of the preceding content is needed.

Line 301-302 It may be a matter of language, but this statement is too strong. Eqs (12,13,14) are new indeed, but they may be considered as another way of expressing the thrust. Methods used so far do not use $P_D$ or $(a^2\text{-}v^2)$ but express T in terms of the rotational speed or lambda and the circulation or swirl around the axis, see eq. 4.6 in Sørensen(2016) and 4.23 in vanKuik(2018).

Our claim is to have quantified the effect of the pressure on the bounding streamsurface on the thrust of an actuator disc. Whether the effects of $a^2\text{-}v^2$ are absorbed in other formulae is not at issue; for example Equation (1) is exact but does not include $a^2\text{-}v^2$. This is one of several good reasons to use the KJ equation rather than the conventional axial momentum equation.

Line 332/333: this statement is not correct. The 'common derivation' of the B-J performance is the derivation for infinite lambda. It does not need information about the distribution of the axial and radial velocity at the disc, nor does it predict the distribution. So what you use 'ignores' I would say 'does not need'. Things are different when you consider annuli instead of the entire stream-tube. Indeed no closed form solution is available, but an analysis (with calculated pressure) is as mentioned in my remark on lines 28-30.

We disagree with this assessment. None of the common derivations of the BJ limit include the $a^2$ and $v^2$ terms in their equivalent of the new Equations (29) and (30). Since we have seen no justification for their omission, it is reasonable to say they have been ignored or not included due to ignorance.

Finally: Abstract and Conclusions : I expect that the abstract and the conclusions will be modified when (some of) my remarks have been taking into account.

All sections of the manuscript have been extensively modified in response to the reviewer comments, to developments in our own thinking, and to correct minor mathematical errors. Changes are shown in blue.

---

## Author Comment (AC2)

The authors present expression of the "general" momentum theory of an actuator disk (including the pressure terms acting on the control volume surfaces, and without assuming that the pressure is recovered in the far wake). In their analyses, they relate the contribution of the pressure term to an integral over the axial and radial momentum in the radial direction. The formulae are presented in integral and differential form. The authors then proceed to studying a finitely bladed rotor with expanding wake but constant pitch.

The work is highly relevant and thorough. I have several general comments that I hope can improve the paper:

 - I would advise to split this paper into two. The actuator disc and finite number of blades part are somewhat related, but each part could very well be put into separate, shorter papers. Mixing disc and finitely-bladed rotors adds in complexity and can confuse the reader.

We have clarified the split between infinite and finite number of blades in response to the first reviewer.  We appreciate that the paper is long but hope it is sufficiently unified to justify keeping together.  There is a practical issue in that the grant that the remaining funds for this work are insufficient to pay for a second paper.

 - The paper contains a lot of maths and is not easy to follow without a definite engagement from the reader. I would recommend to guide the reader more between equations: stating what equations are used, adding intermediate steps and definitions, translating definitions into maths, etc. I think it would help the reader, if going from one equation to the next is straightforward. I've added specific comments in the pdf for equations that in my opinion need more guidance.

I enclose some specific comments in the pdf attached to this review. I'd like to congratulate the authors for their interesting work. I'll be looking forward to review a revised version of this paper.

(combined answer to last two queries) We have looked at all the annotations on the pdf in making our revisions and list our detailed responses below.

Page 2: Is there an assumption related to the pressure on the control volume boundary? Could you mention why no pressure term is present in this equation? (I apologize, I haven't reviewed your previous work yet).

Could you briefly mention in the text what was used to derive this formula? (conservation of axial momentum and some angular momentum considerations?)

(combined answer to last two queries) As noted in the text the impulse equations for force have had the pressure removed in their basic formulation.  In the paragraph preceding equation (1), we have added a sentence and a reference to Noca's thesis (1997) to highlight that the pressure removal is done by substitution of various identities into a standard momentum CV analysis.  For more details, see LW or Noca (1997).  In response to the following comments about pressure, we likewise point the reviewer to LW.  LW

mention that the impulse formulation appears to give no benefit in analyzing angular momentum but this equation is not used in the current manuscript.

Could you describe "circumferential" using a coordinate system to avoid any confusion on what is meant here?

We have added text to clarify the definition of the circumferential co-ordinate. We use x instead of r for consistency with LW.

I would suggest using r instead of x.

We have kept the notation used in LW to make it easier for readers to switch from that paper to this one.

Page 3: Are you using an actuator disk assumption? You mention blades here, are they lifting lines? It seems like assumptions on the rotor loading need to be stated.

We have clarified the assumptions relating to the actuator disc. Equations (1) – (3) are applicable to an actuator disc.

the choice of u and v does not really match the convention where "u" is along "x" (since you chose x along r).

As explained above, we have kept the notation used in LW to make it easier for readers to switch from that paper to this one.

Could you justify assumption 8?

LW show that this assumption is required to recover the Kutta-Joukowsky expression for local thrust that is conventionally employed in blade-element momentum analyses. A note to this effect has been added in the paragraph after the list of assumptions. To remove the assumption would require a model for the vorticity crossing the other faces, which we do not have.

Is there an assumption relating the pressure in the far wake (recovered and equal to free stream)? Coming back up, it seems you don't, since you keep Pinfty.

We do not assume equality of free-stream and far-wake pressure – they differ because of the new Equation (19). The situation has been clarified by adding that we treat all pressures as gauge pressures relative to the free-stream.

It seems you use non dimensionalized velocities, I don't think it was mentioned above (I might have missed it).

Thanks for pointing this out. The original text moved from dimensional to non-dimensional quantities without comment. The text has been revised.

Page 5: It seems to me that an assumption on the pressure on the side of the control volume (at radius=infinity and infinity upstream) needs to be mentioned here.

Please see the comments above about pressure.

Also, P_D seem to be be the pressure "minus" the infinity upstream pressure P_0. This could be precised in the text for clarity.

You are correct and the text has been modified to indicate that all pressures are gauge pressures.

It might be worth stating in the text what this means (there is no pressure jump outside ofthe actuator disk)

A sentence to that effect has been added immediately below Equation (16).

I believe BS ends at z=0+, could you mention this in the text?

No, the BS extends to the far-wake but its involvement with the thrust is determined by consideration of the CVs in Figure 1, which end in the immediate vicinity of the rotor.

Could you detail how the normal to the streamtube projected on the axial direction is obtained here?

This has been done.

Page 6: Can you justify why this generalization is possible. I'm guessing it comes from the fact that all the equations used are valid upstream (with T=0). But this doesn't appear obvious straightaway to the reader since you made use of S_D multiple times in the developments above.

You are correct and a phrase has been added about T = 0.

Page 7: Can you introduce a couple of temporary steps here, showing first an equation similar to (12), and then showing how the terms are expressed? It is not straightforward to me to see how the term int_-inf^inf Pdx/dz xdz  term got manipulated here.

The first part of the new Equation (22) is just a standard application of the axial momentum balance to a CV enclosing the far-wake. The second part uses (19) to remove the pressure. We do not think that additional justification is needed for these well-known results.

It this derived from equation (22), can you make the assumptions more explicit so that the reader can easily go from 22 to 23?

The new (24) comes from the full form of the impulse version of the T equation.  The appropriate reference to the equation in LW has been added.

Intermediate steps/ helps would be at here again to guide the reader and see how u and Delta P where introduced based on the variables in the bracket.

The description of the steps in obtaining (27) from (26) have been expanded.

This statement holds out outside and inside the wake? or only outside of the wake region? Could you please precise?

As noted above, all the discussion that uses the term "redistribution" has been revised.

Page 18: I'm guessing you are doing the theta integration here to find the average, is this correct? can you precise in the text?

We hold that the description of the process is sufficient. Immediately below the new (41) we explain that the sin terms vanish on circumferential integration and we have carefully noted that the arguments for $i_a$ etc show whether they are for velocities at a point or are circumferential averages. Further, section 4.1 starts by stating the goal of obtaining the circumferential averages.

---

## Referee Report (RR1)

Review of the revised version of:

**Some effects of flow expansion on the aerodynamics of horizontal axis wind turbines**

by David H. Wood and Eric J. Limacher

file number WES-2021-52-ATC1

Reviewer: Gijs van Kuik, TU-Delft

The paper has been modified by the authors as a response to the reviews, with an explanation of the changes by the response of the authors to the two reviewers. The resulting version is almost ready for publication, only 1 topic requires a correction: Table 1. As in the first version of the manuscript, $\Delta C_T$ is not the difference between $C_T$ and $C'_T$. Furthermore the explanation of $C_T$ and $C'_T$ is given only in lines 384-391, while it is better placed where Table 1 is first mentioned, so in the beginning of section 4.2.

Apart from this, the manuscript is ready. This does not imply that I share all explanations and interpretations, but that is not part of a review as the paper is clear (given that it is very mathematical) and sound. Having different views on explanations and interpretations drives scientific discussions, but does not prohibit a positive review.

20-9-2021

---

## Author Response (AR2)

Review of the revised version of:

**Some effects of flow expansion on the aerodynamics of horizontal axis wind turbines** by David H. Wood and Eric J. Limacher file number WES-2021-52-ATC1

Reviewer: Gijs van Kuik, TU-Delft

The paper has been modified by the authors as a response to the reviews, with an explanation of the changes by the response of the authors to the two reviewers. The resulting version is almost ready for publication, only 1 topic requires a correction: Table 1. As in the first version of the manuscript, $\Delta C_T$ is not the difference between $C_T$ and $C'_T$ . Furthermore the explanation of $C_T$ and $C'_T$ is given only in lines 384-391, while it is better placed where Table 1 is first mentioned, so in the beginning of section 4.2.

Thanks for these comments. The values in the Table have been corrected and the table and text have been re-ordered to improve the placement and order of the discussion. The captions to figures 2-5 have been reviewed and improved. In addition, a few minor typographical and other errors were removed.

Apart from this, the manuscript is ready. This does not imply that I share all explanations and interpretations, but that is not part of a review as the paper is clear (given that it is very mathematical) and sound. Having different views on explanations and interpretations drives scientific discussions, but does not prohibit a positive review.

We appreciate these comments and look forward to further discussions.

Dear authors,

Thank you for your work and addressing my comments.

I would like to suggest to add something in the text regarding two small questions I had for your original manuscript. The reason I'm mentioning them again is that those points confused me, so I believe they might confuse other readers. I would suggest to add some clarifications/intermediate steps. My two comments were the following:

- I believe BS ends at z=0+, could you mention this in the text?

The upper limits in equations (13) and (14) have been changed to 0+, along with an explanatory note in the text just before equation (13). The location +0 has also been mentioned in the last sentence of the penultimate paragraph in Section 2.

- Page 7 (of original manuscript): Can you introduce a couple of temporary steps here, showing first an equation similar to (12), and then showing how the terms are expressed? It is not straightforward to me to see how the term int_-inf^inf Pdx/dz xdz term got manipulated here.

In fact, we have not evaluated the integral at all. We simply equate the last terms on the right-hand sides of (12) and (13), since all remaining terms in those equations are identical. This explanation has been added after equation (13).

I consider these optional and don't want to appear pushy, I just hope they can help other readers.